# Training-Trajectory-Aware Token Selection

**Zhanming Shen** [1 2]   **Jiaqi Hu** [1 2]   **Zeyu Qin** [3]   **Hao Chen** [1]   **Wentao Ye** [1 2]   **Zenan Huang** [2]   **Yihong Zhuang** [2]
**Guoshan Lu** [2]   **Junlin Zhou** [2]   **Junbo Zhao** [1 2]

## Abstract

Efficient distillation is a key pathway for converting expensive reasoning capability into deployable efficiency, yet in the frontier regime where the student already has strong reasoning ability, naive continual distillation often yields limited gains or even degradation. We observe a characteristic training phenomenon: even as loss decreases monotonically, all performance metrics can drop sharply at almost the same bottleneck, before gradually recovering. We further uncover a token-level mechanism: confidence bifurcates into steadily increasing **Imitation-Anchor Tokens** that quickly **anchor** optimization and other yet-to-learn tokens whose confidence is suppressed until after the bottleneck. And the characteristic that these two types of tokens cannot coexist is the root cause of the failure in continual distillation. To this end, we propose **Training-Trajectory-Aware Token Selection (T3S)** to reconstruct the training objective at the token level, clearing the optimization path for yet-to-learn tokens. T3S yields consistent gains in both AR and dLLM settings: with only **hundreds** of examples, Qwen3-8B surpasses **DeepSeek-R1** on competitive reasoning benchmarks, Qwen3-32B approaches **Qwen3-235B**, and T3-trained LLaDA-2.0-Mini exceeds its AR baseline, achieving **state-of-the-art performance among all of 16B-scale no-think models**.

## 1. Introduction

As large language models (LLMs) with long chain-of-thought (CoT) capabilities continue to emerge (Jaech et al., 2024; Yang et al., 2025; Guo et al., 2025), reasoning distil-

lation is becoming a key pathway for converting expensive reasoning ability into deployable efficiency (Luo et al., 2025; Qin et al., 2025; Guo et al., 2025). Recent work increasingly points to **efficient distillation** as the key lever: rather than scaling data volume, high-quality distilled trajectories and carefully selected examples can produce meaningful reasoning gains with only hundreds to thousands of samples (Muennighoff et al., 2025; Ye et al., 2025b; Wu et al., 2025).

However, in the **frontier regime** where the student already has strong reasoning ability, it remains unclear how to continually push performance via distillation. Prior work and our revisits suggest that naive continual distillation often yields limited gains, and distilling from an out-of-distribution (OOD) teacher can even cause severe degradation (Shen et al., 2025b). This raises a basic question: *what goes wrong during continual reasoning distillation?*

To answer this, we log every checkpoint during distillation. While training loss decreases monotonically, all performance metrics show the same trajectory: **performance drops sharply to an almost fixed minimum stage and then gradually recovers**. We term this **Imitation Shock** and call the checkpoint with the lowest training accuracy the **Imitation Bottleneck**. Our post-bottleneck recovering-residual transfer test shows the pre-bottleneck phase is not required: discarding pre-bottleneck updates can improve generalization, implying that early updates may be harmful.

We then probe token-level dynamics and uncover a compact explanation. Across checkpoints, token confidence separates into two groups: one group increases steadily, while the other decreases steadily, and this split persists until the Imitation Bottleneck; only after the bottleneck do both groups start improving (Figures 4). We call the first group **Imitation-Anchor Tokens**. We find that the root cause of this dynamics is that learning Imitation-Anchor Tokens exerts a strong suppressive effect on other yet-to-learn tokens; this suppression weakens only after anchor tokens have been learned to a sufficient extent (Figures 5). We view this anchor-induced suppression as the underlying driver of Imitation Shock. In short, **direct continual distillation fails not because reasoning is absent, but because it is systematically delayed by imitation anchors**.

Moreover, we even find that the gradients of anchor to-

[1]Zhejiang University [2]Inclusion AI, Ant Group [3]Hong Kong University of Science and Technology. Correspondence to: Zhanming Shen <z.shen@zju.edu.cn>, Junbo Zhao <j.zhao@zju.edu.cn>.

*Proceedings of the 43rd International Conference on Machine Learning*, Seoul, South Korea. PMLR 306, 2026. Copyright 2026 by the author(s).

kens and other yet-to-learn tokens could not **coexist benignly**—progress on one side tends to come at the expense of the other (Table 6). Together, these observations motivate **Training-Trajectory-Aware Token Selection (T3S)**: use the training-trajectory signal to find the Imitation Bottleneck and reconstruct training at the token level, preventing early **Imitation-Anchor Tokens** from dominating optimization and delaying reasoning transfer, clearing the optimization path for **yet-to-learn tokens**.

Empirically, T3S yields large gains across both autoregressive (AR) model and diffusion language model (dLLM) settings. With only **hundreds of** training examples, in the AR regime Qwen3-8B surpasses its teacher DeepSeek-R1 on competitive reasoning benchmarks, and Qwen3-32B even achieves results comparable to the Qwen3-235B model. In the dLLM regime, T3S-trained LLaDA-2.0-Mini substantially improves reasoning accuracy and surpasses its shared-architecture AR baseline, achieving **state-of-the-art performance among all of the 16B-scale no-think models**.

**Contributions.** We make three contributions:
(1) We identify **Imitation Shock** as a pervasive phenomenon in continual reasoning distillation and show that pre-bottleneck updates (the Imitation-Anchor Tokens) are not required and can be harmful for yet-to-learn tokens.
(2) We propose **T3S** for AR models by masking **Imitation-Anchor Tokens** to clear the optimization path for yet-to-learn tokens, yielding consistent gains under various settings.
(3) We extend T3S to **dLLMs** by repeatedly practicing yet-to-learn tokens via AR selector and union masking, improving both reasoning accuracy and efficiency.

**Conflict of Interest Disclosure.** All the authors belonging to Inclusion AI leads the development of LLADA 2.0, which was among the ones evaluated in this paper.

## 2. Imitation Shock Enables Targeting

### 2.1. Revisiting distillation dynamics

We revisit a canonical efficient distillation setting where Qwen3-8B is distilled from DeepSeek-R1, motivated by a **recurring failure mode of direct continual distillation**: despite optimizing the distillation objective, model performance often yields limited net gains or even degradation (See Figure 1). We use a batch size of 64, a learning rate of $1 \times 10^{-5}$, and train 200 samples from BOBA-200 (Shen et al., 2025b; inclusionAI, 2025) for 50 optimization steps. We save *every* checkpoint and measure not only training loss but also training-set answer accuracy and multiple downstream benchmarks. Surprisingly, while the training loss decreases monotonically, all evaluated metrics, including AIME24 (Math-AI, 2024), AIME25 (Math-AI, 2025), MMLU-Pro (Wang et al., 2024), and even the training-set

answer accuracy, exhibit an almost identical pattern: **performance first drops sharply at almost the same stage and then gradually recovers**. We term this phenomenon **Imitation Shock** and refer to the checkpoint attaining the lowest training accuracy as the **Imitation Bottleneck**. Figure 1 illustrates this behavior under DeepSeek-R1 distillation on BOBA-200. Beyond this case, we find that the same "crash then recover" pattern persists across substantially broader settings—different teachers, datasets, dataset scales, student models, and even training domains (Figure 2); detailed configurations are provided in Appendix E. These observations suggest that **the degradation in direct continual distillation is not incidental**: it follows a characteristic "crash then recover" trajectory that persists across diverse settings—spanning different teacher–student pairs, datasets, dataset scales, and even training domains.

*Takeaway 1. In continual distillation, loss can fall while performance crashes then recovers; We term this phenomenon Imitation Shock.*

### 2.2. Recovering-Residual Transfer from the Imitation Bottleneck

**Motivation: is pre-bottleneck training necessary?** The emergence of **Imitation Shock** raises a natural question about the role of pre-bottleneck updates. Since model performance drops sharply and then recovers, it is unclear whether the optimization trajectory before the **Imitation Bottleneck** represents a necessary phase of learning or merely a transient detour during teacher–student alignment. Motivated by this observation, we ask a simple but fundamental question: *can the student benefit from distillation without going through the pre-bottleneck phase at all?*

**Recovering-residual transfer construction.** To answer this question, we explicitly remove all parameter updates before the Imitation Bottleneck and retain only the updates acquired afterward. Let $\theta_0$ denote the base initialization, $\theta_b$ the bottleneck checkpoint (defined as the point of lowest validation accuracy), and $\theta_f$ the final distilled model. We define a **Recovering Residual Transfer (RRT)** as $\Delta\theta_{\mathrm{RRT}} = \theta_f - \theta_b$ and construct a new model $\theta_{\mathrm{RRT}} = \theta_0 + \Delta\theta_{\mathrm{RRT}}$, which entirely ignores all pre-bottleneck updates.

**Results and implications.** As shown in Table 1, models constructed using only post-bottleneck updates consistently outperform standard distillation on both BOBA-200 and S1K-200. Remarkably, distillation from DeepSeek-R1, where standard SFT leads to a net performance drop, becomes a substantial gain when only post-bottleneck updates are retained. These results demonstrate that the learning process prior to the Imitation Bottleneck is not required for effective generalization and may even interfere with it. This finding suggests that **early-stage distillation updates encode information that is neither necessary nor beneficial**.

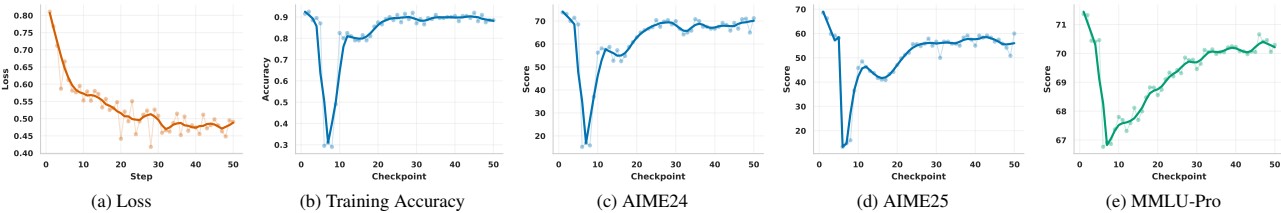

*Figure 1.* **Imitation Shock under DeepSeek-R1 distillation on BOBA-200.** Although training loss decreases monotonically (a), training-set answer accuracy and multiple benchmarks (AIME24/25, MMLU-Pro) drop sharply to a shared minimum stage and then recover, revealing an *Imitation Shock*.

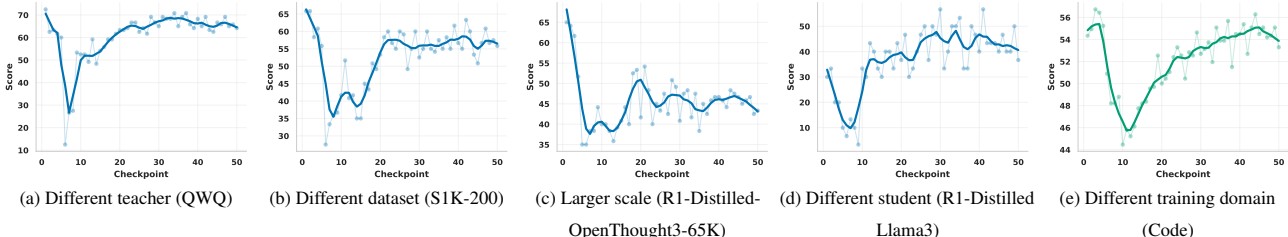

*Figure 2.* **Imitation Shock is universal across settings.** We observe the same "crash then recover" trajectory across (a) different teachers, (b) different datasets, (c) larger-scale datasets, (d) different student backbones, and (e) different training domains. See Appendix E for detailed setups and metrics.

*Table 1.* **Recovering-residual transfer extracted from the Imitation Bottleneck.** $\Delta$Avg denotes the average performance change relative to the BASE model. Standard distillation leads to degradation, while retaining only the post-bottleneck updates converts the drop into consistent gains.

| Method | BOBA-200 | | | S1K-200 | | |
|--------|----------|--------|-------|---------|--------|-------|
| | AIME24 | AIME25 | $\Delta$Avg | AIME24 | AIME25 | $\Delta$Avg |
| BASE | 75.83 | 67.08 | – | 75.83 | 67.08 | – |
| SFT (R1) | 71.25 | 55.00 | ↓8.33 | 72.50 | 55.84 | ↓7.29 |
| RRT (R1) | 76.67 | 68.54 | ↑1.15 | 76.67 | 70.63 | ↑2.20 |
| SFT (QWQ) | 73.33 | 63.33 | ↓3.13 | 73.33 | 64.17 | ↓2.70 |
| RRT (QWQ) | 76.04 | 69.17 | ↑1.15 | 75.00 | 71.67 | ↑1.88 |

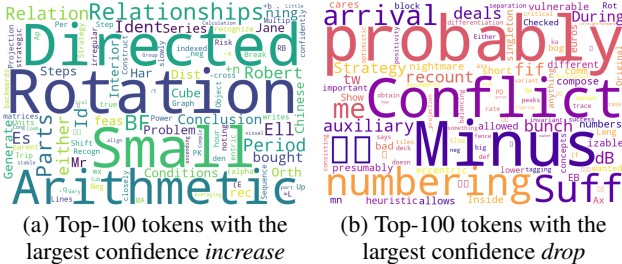

(a) Top-100 tokens with the largest confidence *increase*

(b) Top-100 tokens with the largest confidence *drop*

*Figure 3.* Word clouds at the Imitation Bottleneck (identified by training-set accuracy). Token sizes are proportional to the magnitude of confidence change relative to the base model.

**Crucially, it motivates us to attribute the distillation trajectory by jointly tracking model-side dynamics and data-side dynamics: what** parts of the teacher traces are being learned before versus after the bottleneck, **why** the pre-bottleneck phase induces degradation, and **how** we can operationalize these dynamics into an observable signal. In the next section, we therefore move from parameter-level analysis to **token-level** attribution, using token confidence changes to reveal which tokens dominate early optimization and which tokens carry delayed reasoning transfer.

*Takeaway 2. Pre-bottleneck updates are not necessary for effective continual distillation; keeping only post-bottleneck updates works better.*

### 2.3. Anchor Tokens and Persistent Learning Barriers

**Token-level evidence at the Imitation Bottleneck.** To further probe what parts of the teacher traces are being learned

before versus after the bottleneck, we inspect token-level statistics at the imitation bottleneck checkpoint identified by training-set accuracy and rank tokens by their confidence change relative to the base model. We find a clear split at the bottleneck stage: **some tokens experience severe confidence drops and become harder to fit**, while **others exhibit large confidence gains and become easier to fit**. To make this trajectory signal tangible, we visualize the **top-100 tokens with the largest confidence increase** and the **top-100 tokens with the largest confidence drop** (Figure 3). Importantly, this bottleneck-referenced confidence change provides a simple and concrete way to **select tokens directly from the distillation trajectory**, which we will use in the following sections to define trajectory-aware token sets based on $\Delta c_t$ (equivalently, $\Delta$probability).

**Persistent token-level learning barriers under prolonged training.** Motivated by the bottleneck analysis, we next

*Table 2.* **Persistent token-level confidence drops relative to the base model under prolonged training.** Results are measured at the final checkpoint after training on ∼200 examples for more than 15 epochs with batch size 64.

| Dataset | Tokens with Confidence Drop (%) |
|---------|--------------------------------|
| BOBA-200 | 68.51 |
| S1K-200 | 53.03 |

ask whether the hard-to-fit tokens (those with confidence drops) eventually recover if we simply train longer. We revisit the same distillation setting under an extended training regime to characterize token-level learning dynamics. As summarized in Table 2, even when training on fewer than 200 examples for over 15 epochs, a large fraction of tokens still exhibit **lower confidence than the base model**. These results indicate that, even under prolonged training, more than half of the tokens are not merely insufficiently learned but are learned **worse** than in the base model. This suggests that learning dynamics at the token level are likely coupled: **the learning of some tokens may interfere with the learning of others**, such that progress on one subset of tokens comes at the expense of another. In particular, the persistence of confidence drops suggests that the issue is not only that some tokens are hard to learn, but that the tokens that become easy to learn early in training may actively dominate optimization and hinder recovery of the hard-to-fit ones.

**Anchor tokens and delayed acquisition of reasoning tokens.** Using the Imitation Bottleneck as a temporal reference, we analyze token-wise learning dynamics on BOBA-200 by tracking probability changes relative to the base model across training checkpoints. As shown in Figure 4(a), when training on the full distillation objective, token confidence separates into two distinct behaviors: **one group increases steadily from early checkpoints, while the other group drops early and only starts to recover after the former group has largely stabilized**. Notably, the recovery of the latter group begins almost exactly when the **Imitation Bottleneck** emerges, suggesting that we may have identified the token-level origin of Imitation Shock.

To test whether this delayed trajectory reflects a *coupled* learning process, we further consider an intervention that **trains only the latter group** (i.e., excludes the steadily-increasing tokens from the loss). Strikingly, under this "train other tokens only" setting (Figure 4(b)), the previously delayed tokens exhibit an almost **opposite** trend: basically, their confidence rises **monotonically** throughout training, rather than crashing first. This contrast suggests that the early dominance of the steadily-increasing group is not merely correlated with the delayed recovery, but can actively **shape** the learning trajectory of the remaining tokens.

We refer to the steadily-increasing group as **Imitation-**

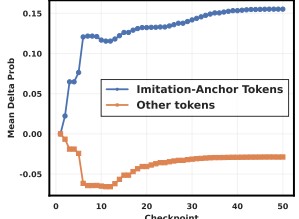
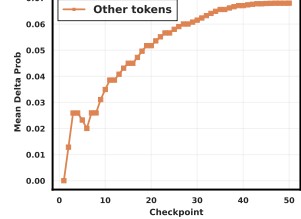

(a) Delta Prob Trends for Two Token Groups **(full objective)**

(b) Delta Prob Trends for Two Token Groups **(train other tokens only)**

*Figure 4.* **Token-wise learning dynamics on BOBA-200.** (a) Under the full distillation objective, anchor tokens increase monotonically, while the remaining tokens crash early and recover only after anchors stabilize. (b) When training only the remaining tokens (excluding anchor tokens from the loss), their confidence rises monotonically, reversing the "crash-then-recover" pattern and indicating suppressive coupling from anchor-token learning.

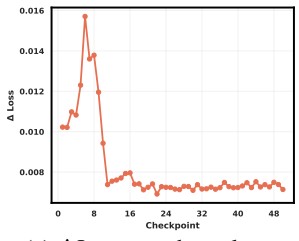
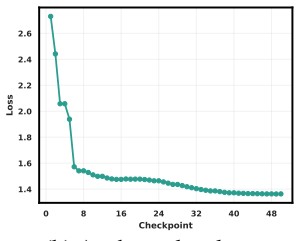

(a) ΔLoss on other tokens

(b) Anchor-token loss

*Figure 5.* **One-step intervention on Imitation-Anchor Tokens.** Panel (a) reports $\Delta\mathcal{L}_{\text{other}}$ after one gradient step that optimizes only anchor tokens at each checkpoint. Panel (b) shows the corresponding anchor-token loss $\mathcal{L}_{\text{anchor}}$ at that checkpoint.

**Anchor Tokens**: they quickly **anchor** optimization and impede the learning of the remaining tokens that are more beneficial for downstream reasoning. Together, these observations provide a training trajectory-level account of why direct continual distillation can degrade performance: early updates are dominated by Imitation-Anchor Tokens, systematically delaying the acquisition of the tokens that matter for generalization.

**Causal evidence for anchor-induced suppression.** To directly test whether Imitation-Anchor Tokens causally interfere with learning other tokens, we perform a checkpoint-wise intervention on the original SFT trajectory. For each saved checkpoint $\theta^{(k)}$, we take *one* gradient step that optimizes *only* Imitation-Anchor Tokens (i.e., we backpropagate loss on anchor positions only), producing an updated checkpoint $\tilde{\theta}^{(k)}$. We then measure how this anchor-only step changes the loss on *all other tokens* (non-anchor positions), using $\Delta\mathcal{L}_{\text{other}}^{(k)} = \mathcal{L}_{\text{other}}(\tilde{\theta}^{(k)}) - \mathcal{L}_{\text{other}}(\theta^{(k)})$, and simultaneously log the anchor loss $\mathcal{L}_{\text{anchor}}(\theta^{(k)})$ across checkpoints. As shown in Figure 5, when anchor tokens are not yet learned (large $\mathcal{L}_{\text{anchor}}$; panel (b)), a single anchor-only step induces a large increase in non-anchor loss (positive

$\Delta \mathcal{L}_{\text{other}}$; panel (a)), indicating strong suppression on other tokens. As training progresses, $\mathcal{L}_{\text{anchor}}$ drops rapidly and the suppression effect decays and stabilizes, after which non-anchor tokens can be learned more steadily. Together, this intervention provides mechanistic evidence that anchor tokens dominate early optimization and can actively inhibit the learning of reasoning-relevant tokens.

**Other Gradient Dynamics Between Token Groups.** To further support the mechanistic claim that imitation anchors suppress the remaining tokens, we track the gradient norms and gradient directions of the two token groups across checkpoints. As shown in Figure 8(a), in early training the gradient norm on Anchor tokens can be overwhelmingly larger (up to $17\times$) than that on Other tokens, indicating that anchors dominate the optimization bandwidth. As anchor loss rapidly decreases, this norm ratio contracts and reaches approximately $2\times$ around the Imitation Bottleneck, at which point Other-token gradients begin to acquire a sufficient "voice" and model performance starts recovering. Importantly, Anchor-token gradients remain larger throughout training, suggesting that without masking, anchors continue to dominate updates. We also compute the cosine similarity between the two group gradients to characterize directional interference. As shown in Figure 8(b), during the crash phase the cosine similarity drops to roughly $-0.4$ to $-0.5$, indicating strong conflict between the two groups. Right after the bottleneck it briefly rises (around $-0.1$) during rapid recovery, and then settles back to a consistently negative regime. These dynamics provide a gradient-level view consistent with our loss-transfer asymmetries (Table 6) and further motivate a hard mask: the two groups' gradients do not coexist benignly, so suppressing anchor gradients directly clears the optimization path for yet-to-learn tokens.

***Takeaway 3.*** *Direct continual distillation fails not because reasoning is absent, but because it is systematically delayed by imitation anchors.*

### 2.4. Why Training-Trajectory-Aware selection is necessary.

Taken together, our evidence indicates that the failure of direct continual distillation is driven by a **token-level incompatibility** that cannot be resolved by naive optimization. First, Figure 4 and Figure 5 show that once **Imitation-Anchor Tokens** begin to dominate learning, the remaining tokens are pushed into a delayed "crash-then-recover" regime, and removing anchors from the objective flips the trajectory into a largely monotonic improvement, revealing a strong **suppressive coupling** from anchor-token learning. Second, the Recovering-Residual Transfer (RRT) result in Section 2.2 demonstrates that the updates acquired *before* the Imitation Bottleneck are not only unnecessary but actively harmful: discarding them converts the performance

crash into consistent gains, implying that what is learned in the pre-bottleneck phase (where anchors dominate) is primarily **net-negative** for downstream performance.

Finally, Table 6 makes this incompatibility explicit: We split tokens within each group by base-model confidence into two equal-sized halves. Training on anchor subsets reduces anchor losses but increases losses on reasoning subsets (often dramatically), while training on reasoning subsets improves reasoning losses but degrades anchor losses. This sharp $4 \times 4$ asymmetry, together with Figure 8, indicate that anchor tokens and other yet-to-learn tokens form a highly **separable two-way partition** whose gradients do not **coexist benignly**—progress on one side tends to come at the expense of the other.

Therefore, for effective distillation we must **eliminate anchor-token gradient dominance** and actively **prioritize the other yet-to-learn tokens** that are otherwise suppressed during early optimization. This is precisely what T3S achieves: by using trajectory signals around the bottleneck to identify and mask anchors early, it **clears the optimization path for other tokens to be learned first**, thereby preventing anchor domination and accelerating the transfer of useful reasoning behavior.

***Takeaway 4.*** *Effective continual distillation requires clearing the optimization path for yet-to-learn tokens.*

## 3. Method

### 3.1. Overview

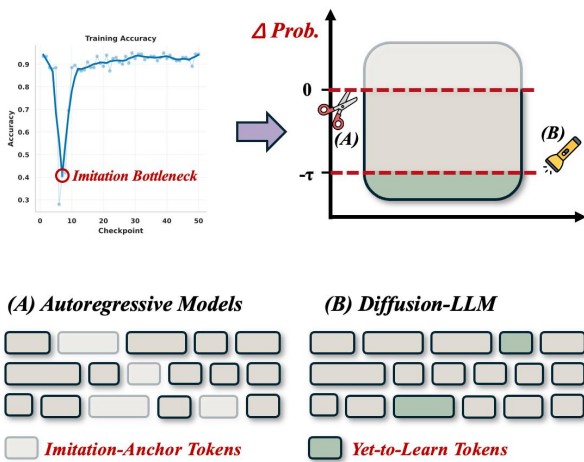

**(A) Autoregressive Models**     **(B) Diffusion-LLM**

*Imitation-Anchor Tokens*     *Yet-to-Learn Tokens*

*Figure 6.* Workflow of our method.

As shown in Figure 6, T3S uses the training trajectory around the **Imitation Bottleneck** to profile token-wise confidence changes and derive token-level targets. For autoregressive (AR) distillation, we identify **Imitation-Anchor tokens** that are learned early during Imitation Shock and mask them from the AR loss, thereby concentrating optimization on the remaining yet-to-learn tokens. For diffusion-

style language models (dLLMs), the training objective is intrinsically random-masked reconstruction; thus, we instead preferentially mask trajectory-identified yet-to-learn tokens so the model repeatedly practices generating them under arbitrary visible-token conditions. We present the AR formulation in the main text, and defer the full dLLM instantiation and analyses to Appendix D.

### 3.2. Imitation Shock and the Imitation Bottleneck

**Imitation Shock**. In continual distillation, we observe a characteristic learning curve where the student's training accuracy drops sharply early in training and then slowly recovers; we call this phenomenon *Imitation Shock*.

**Imitation Bottleneck.** Let $\text{Acc}_{\text{train}}(\theta)$ denote t raining accuracy of the student model at parameters $\theta$. We define the *Imitation Bottleneck* checkpoint as

$$\theta_b = \arg \min_{\theta \in \{\theta^{(0)}, \theta^{(1)}, \dots\}} \text{Acc}_{\text{train}}(\theta), \quad (1)$$

i.e., the checkpoint with the **lowest training accuracy** along the distillation trajectory. Let $\theta_0$ be the starting checkpoint (pre-distillation initialization).

T3S uses the training-trajectory signal around $\theta_b$ to construct token-level targets for (i) AR distillation and (ii) dLLM training, with different token selections for each.

### 3.3. Training-Trajectory-Aware Token Confidence Profiling

**Token confidence.** We use an AR token selector model $M_0$ to score token-wise confidence along the training trajectory. For any $(x, y)$ and position $t$, define

$$c_t(\theta; x, y) = \log p_\theta(y_t \mid y_{<t}, x). \quad (2)$$

We measure the trajectory change from $\theta_0$ to the Imitation Bottleneck $\theta_b$:

$$\Delta c_t(x, y) = c_t(\theta_b; x, y) - c_t(\theta_0; x, y). \quad (3)$$

### 3.4. T3S for AR Distillation: Masking Imitation-Anchor Tokens

For AR distillation, our goal is to **mask tokens that the student becomes confident about already during Imitation Shock**, which help the student rapidly "lock onto" imitation but may not yield commensurate reasoning gains.

**Imitation-Anchor Tokens (confidence-increase).** We refer to **Imitation-Anchor Tokens** as the subset of tokens that the student aligns with first during **Imitation Shock**; they quickly **anchor** optimization, but do not necessarily contribute to improved reasoning performance and even exert

a strong suppressive effect on other tokens;. Operationally, we define

$$\mathcal{A}(x, y) = \{t \in [T] \mid \Delta c_t(x, y) > 0\}. \quad (4)$$

**Masked AR objective.** We **mask** $\mathcal{A}(x, y)$ by excluding them from the AR loss (equivalently, setting zero weight):

$$\mathcal{L}_{\text{AR-T3S}} = \mathbb{E}_{x,y} \left[ \sum_{t \backslash \mathcal{A}(x,y)} -\log p_\theta(y_t \mid y_{<t}, x) \right]. \quad (5)$$

This objective discourages over-optimizing tokens that are absorbed early during Imitation Shock, focusing capacity on the remaining positions that are more beneficial for downstream reasoning.

### 3.5. Practical Considerations and Overhead

**Compute cost.** In practice, T3S measurements can be performed online during training. We log and evaluate every checkpoint mainly for experimental analysis. A practical implementation can **mirror early stopping** in standard Transformer pipelines: monitor the relevant metrics at each update (or periodically) to identify $\theta_b$ on the fly and compute $c_t(\theta)$ with minimal overhead.

**On requiring training accuracy.** Our bottleneck selection uses training accuracy, which assumes access to gold answers (or an automatic verifier) to determine correctness. We do not view this as a major limitation in practice: it is already satisfied by many distillation/RL pipelines. In particular, **any dataset that supports reinforcement learning** via a reward model or outcome-based verification (i.e., provides a reliable correctness metric) can be directly used to instantiate $\text{Acc}_{\text{train}}$ for T3S.

## 4. Experiment

### 4.1. Main Results

**Experimental setup.** Just as previous studies on efficient distillation (Shen et al., 2025b), we use two high-quality open-source mathematical reasoning datasets for training: BOBA-200 (inclusionAI, 2025) and S1K-200 (Muennighoff et al., 2025). We distill Qwen3-8B (Yang et al., 2025) from two contrasting teachers, DeepSeek-R1 (Guo et al., 2025) and QWQ (Team, 2024), by sampling multiple reasoning traces per prompt and training on a randomly chosen correct trace when available. Unless otherwise noted, all methods use the same fine-tuning recipe (batch size 64, learning rate $1 \times 10^{-5}$, 50 steps) and are evaluated on AIME24 (Math-AI, 2024)/AIME25 (Math-AI, 2025) with scores averaged over 16 runs; full details are deferred to Appendix I.

**Compared methods.** We compare four distillation strategies under the same training configuration. **SFT** denotes

standard single-teacher distillation, where the student is directly fine-tuned on reasoning traces generated by a selected teacher. **RRT** corresponds to the preliminary method motivated by our Imitation Shock analysis: we identify the Imitation Bottleneck along the training trajectory and retain only post-bottleneck parameter updates via a recovering residual transfer construction (See subsection 2.2). **T3S (Training-Trajectory-Aware Token Selection)** is our proposed method, which reconstructs the training objective at the token level by explicitly masking imitation anchor tokens during optimization to clear the optimization path for yet-to-learn tokens. Finally, **-T3S** is a diagnostic ablation where we **invert** the T3S mask, i.e., we mask exactly the yet-to-learn tokens identified by T3S and train on the imitation anchor tokens only, to probe the selectivity of T3S.

**Main results.** Table 3 reports the main results on BOBA-200 and S1K-200 under both DeepSeek-R1 and QWQ distillation. Across datasets and teachers, **T3S** consistently achieves the best performance, substantially outperforming standard **SFT** and also improving over **RRT**, which removes pre-bottleneck updates only at the parameter level—highlighting the advantage of reconstructing the objective directly at the **token level** using training-trajectory-aware signals. Moreover, the severe collapse under **-T3S** (masking exactly the tokens selected by T3S) demonstrates that T3S identifies a highly selective and essential token partition rather than discarding redundant positions. Finally, T3S yields larger gains when distilling from **DeepSeek-R1** than from **QWQ**, suggesting that stronger teachers may provide richer signals but also stronger harmful biases under naive distillation; T3S mitigates these biases and unlocks the teacher's latent benefit.

**Beyond AIME: General Reasoning and Forgetting Mitigation.** We further evaluate whether T3S transfers beyond math-centric AIME-style testing. Across multiple out-of-distribution reasoning benchmarks and a set of instruction-following / knowledge-style benchmarks, T3S consistently improves general reasoning and mitigates catastrophic forgetting compared to standard SFT; detailed results are provided in Appendix G.

**Additional generality and comparisons.** Beyond the main BOBA-200/S1K-200 results, we further validate T3S from four complementary angles. (1) **Scale-up:** T3S remains effective when substantially increasing the distillation corpus size, preventing severe degradation under an OOD teacher and continuing to improve under massive in-distribution data (Appendix H.1). (2) **Stronger baselines:** we compare against representative distillation strategies and find T3S consistently yields higher AIME24/25 scores under comparable settings (Appendix H.2). (3) **Cross-family robustness:** applying T3S to a different student architecture still produces substantial gains over SFT, indicating the effect

*Table 3.* **Comparison of distillation methods on BOBA-200 and S1K-200.** We report AIME24, AIME25, and their average (AVG). Results are grouped by (i) base model, (ii) DeepSeek-R1 distillation, and (iii) QWQ distillation.

| Method | BOBA-200 | | | S1K-200 | | |
|---|---|---|---|---|---|---|
| | AM24 | AM25 | AVG | AM24 | AM25 | AVG |
| BASE | 75.83 | 67.08 | 71.46 | 75.83 | 67.08 | 71.46 |
| SFT (R1) | 71.25 | 55.00 | 63.13 | 72.50 | 55.83 | 64.17 |
| RRT (R1) | 76.67 | 68.54 | 72.61 | 76.67 | 70.63 | 73.65 |
| -T3S (R1) | 30.63 | 25.63 | 28.13 | 26.67 | 26.67 | 26.67 |
| **T3S (R1)** | **80.63** | **73.96** | **77.30** | **80.00** | **73.13** | **76.57** |
| SFT (QWQ) | 73.33 | 68.33 | 70.83 | 73.33 | 63.33 | 68.33 |
| RRT (QWQ) | 76.04 | 69.17 | 72.61 | 75.00 | 71.67 | 73.33 |
| -T3S (QWQ) | 50.00 | 33.33 | 41.67 | 46.67 | 40.00 | 43.33 |
| **T3S (QWQ)** | **77.50** | **70.83** | **74.17** | **76.67** | **72.50** | **74.59** |

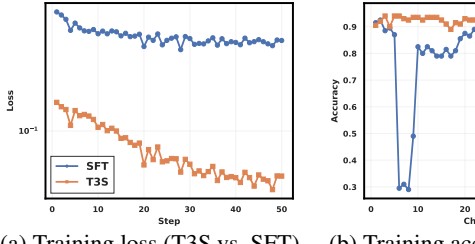

(a) Training loss (T3S vs. SFT)    (b) Training accuracy (T3S vs. SFT)

*Figure 7.* **Training dynamics comparison between T3S and SFT on BOBA-200.**

is not tied to a single model series (Appendix H.3). (4) **Domain transfer:** on a purely code-based dataset, we again observe Imitation Shock and T3S delivers clear improvements on LiveCodeBench, demonstrating that the phenomenon and the fix extend beyond math reasoning (Appendix H.4).

**Training dynamics: T3S vs. SFT on BOBA-200.** To directly connect our mechanism to optimization behavior, we compare the training dynamics of standard SFT and T3S on BOBA-200. As shown in Figure 7(a), after explicitly targeting and masking Imitation-Anchor Tokens, T3S starts from a noticeably lower loss and converges faster. More importantly, Figure 7(b) shows that the characteristic *Imitation Shock* observed in SFT—a sharp accuracy collapse followed by gradual recovery—is largely resolved under T3S: training accuracy no longer crashes and instead improves steadily. Together with our token-level analyses and the downstream gains, these dynamics **complete the logical loop of our hypothesis: early optimization is dominated by imitation anchors, and suppressing them removes the barrier that delays reasoning transfer.**

### 4.2. Debiased Teacher Mixing via T3S

**Motivation and setup.** Having shown that T3S may remove harmful teacher-specific inductive biases at the token level, we next ask whether *mixed* debiased distillation data still

yields gains over standard SFT. To this end, we apply T3S independently to the S1K-200 distillation corpora generated by three teachers with distinct characteristics: DeepSeek-R1, Qwen3-235B (Yang et al., 2025), and QWQ. We then construct mixed training data by aggregating T3S-reconstructed samples from an increasing number of teachers, and evaluate whether the resulting student continues to improve relative to SFT after mixing.

We allocate a fixed budget of 50 optimization steps per teacher, i.e., $50 \times |\mathcal{T}|$ total steps for a mixture with $|\mathcal{T}|$ teachers. We evaluate on two student models, Qwen3-8B and Qwen3-32B (Yang et al., 2025), reporting results on AIME24 and AIME25. As a reference point, we additionally compare against training on T3S-reconstructed DeepSeek-R1 data alone, to verify that teacher mixing preserves (and potentially amplifies) the gains rather than diluting them. For completeness, we also report the base performance of the teacher models themselves.

**Results and analysis.** Several observations emerge from Table 4. **Notably, using only a few hundred training examples, Qwen3-8B exceeds the performance of its teacher DeepSeek-R1 on competitive reasoning benchmarks, and Qwen3-32B achieves results on par with the Qwen3-235B-A22B, despite the large gap in model scale.**

### 4.3. T3S on Diffusion-Style Language Models

Unlike AR models, dLLMs are trained with an intrinsic random-masking objective, learning by reconstructing masked positions under partial observations. Thus, the key question is not which tokens to imitate most aggressively, but which tokens should be **preferentially masked** so the model is forced to learn what it still does not know. We further find that **yet-to-learn** tokens exhibit non-trivial cross-model consistency within tokenizer-sharing model families (Appendix C), suggesting these tokens reflect partially model-agnostic learning barriers rather than purely idiosyncratic failures. Since dLLMs do not provide a clean AR-style token-wise likelihood interface for precise localization, we use an AR selector as a practical proxy to identify yet-to-learn tokens and enforce their inclusion in the dLLM mask via a targeted union-masking strategy (Appendix D).

Table 5 reports our dLLM results. We distill LLaDA-2.0-Mini (Bie et al., 2025) on the S1K dataset; detailed training and evaluation configurations are provided in Appendix D. T3S not only substantially improves dLLM reasoning performance over standard SFT, but also yields a strong standalone dLLM: the T3S-trained LLaDA-2.0-Mini markedly improves reasoning accuracy and surpasses its shared-architecture AR baseline (Ling-mini), achieving state-of-the-art performance among 16B-scale no-think models (Table 5). We further observe that targeted masking improves inference efficiency by concentrating learning

on yet-to-learn tokens (Appendix D.4), and we provide a sensitivity analysis of the key threshold $\tau$ in Appendix D.5.

### 4.4. Robustness to Bottleneck Selection

**A simple heuristic selector.** In our continual distillation setting, we observe that the Imitation Bottleneck tends to occur within a narrow early-stage window (typically between steps 6 and 15). This motivates a simple heuristic variant, **T3S-simple**, which uses a fixed checkpoint (step 10) as the token selector, without explicitly searching for the minimum training accuracy along the trajectory. Despite its simplicity, T3S-simple substantially outperforms direct SFT, and trails only slightly behind the full T3S (Table 7).

**Sensitivity analysis.** To further quantify robustness, we uniformly sample token selectors from checkpoints before and after the true bottleneck and report the final AIME24/25 average. Performance follows a clear inverted-U shape peaking at the true bottleneck, but remains above SFT across a wide range of selector choices (Table 8), indicating that T3S is tolerant to imperfect bottleneck identification. Moreover, using even the very first (ckt 2) or the very last (ckt 50) checkpoint as the selector still outperforms direct SFT. This aligns with our previous analyses: **imitation-anchor tokens dominate optimization from the very beginning, and a large portion of yet-to-learn tokens remains suppressed even at the end of training.** Consequently, both early and late selectors can still successfully target and mask a sufficient number of harmful anchor tokens.

### 4.5. Beyond Initial Confidence

**Goal: separating T3S from static confidence heuristics.** Recent token-selection approaches often rely on *static* signals such as initial uncertainty (token entropy) or base-model confidence (Wang et al., 2025b;a). In **continual reasoning distillation**, however, our central claim is that the token partition that governs optimization is **training-trajectory-defined**: the tokens that dominate early updates cannot be reliably identified from initialization statistics alone. We provide three complementary pieces of evidence: (i) the T3S anchor/non-anchor split is not separable by static confidence or local gradient geometry, (ii) masking the same budget by static confidence fails and **does not remove Imitation Shock**, and (iii) the T3S split corresponds to a functionally meaningful antagonistic partition in training dynamics.

**Non-separability under static confidence and gradient geometry.** The anchor/non-anchor split induced by T3S cannot be reliably recovered from base-model confidence or local gradient geometry: the two groups substantially overlap in both the confidence scatter and the token-gradient sketch visualization (Appendix J). This rules out a purely initialization- or gradient-based explanation for why certain tokens dominate early optimization, and motivates using

*Table 4.* **Debiased teacher mixing with T3S.** We report AIME24, AIME25, and their average (AVG). T3S is applied independently to each teacher's distillation data before mixing.

| Method | Qwen3-8B | | | Qwen3-32B | | |
|---|---|---|---|---|---|---|
| | AIME24 | AIME25 | AVG | AIME24 | AIME25 | AVG |
| Base | 75.83 | 67.08 | 71.46 | 81.46 | 72.08 | 76.77 |
| SFT(R1) | 72.50 | 55.83 | 64.17 | 76.67 | 67.50 | 72.09 |
| SFT(R1+235B) | 75.00 | 64.17 | 69.59 | 81.67 | 66.67 | 74.17 |
| SFT(R1+235B+QWQ) | 73.33 | 65.83 | 69.58 | 78.33 | 69.17 | 73.75 |
| T3S(R1) | 80.00 | 73.13 | 76.57 | 83.33 | 73.33 | 78.33 |
| T3S(R1+235B) | 83.33 | 73.33 | 78.33 | 84.17 | 76.67 | 80.42 |
| **T3S(R1+235B+QWQ)** | **84.17** | **76.67** | **80.42** | **86.67** | **80.00** | **83.34** |
| DeepSeek-R1+Base | 79.80 | 70.00 | 74.90 | 79.80 | 70.00 | 74.90 |
| Qwen3-235B-A22B+Base | 85.70 | 81.50 | 83.60 | 85.70 | 81.50 | 83.60 |

*Table 5.* **Results of T3S on diffusion-style language models.** We report performance on AIME25, MATH500, and TheoremQA, along with their average (AVG).

| Model | AIME25 | MATH500 | TheoremQA | AVG |
|---|---|---|---|---|
| LLaDA-2.0-Mini (Base) | 30.00 | 91.98 | 45.88 | 55.95 |
| LLaDA-2.0-Mini (SFT) | 30.00 | 91.58 | 47.00 | 56.19 |
| **LLaDA-2.0-Mini (T3S)** | **53.33** | **93.19** | 51.50 | **66.01** |
| Qwen3-14B (no-think) | 26.51 | 88.18 | 55.88 | 56.86 |
| Ministral-14B-Instruct-2512 | 48.59 | 91.38 | **56.13** | 65.37 |
| Ling-Mini | 46.25 | **94.79** | 50.38 | 63.81 |
| Qwen3-30B-A3B-Instruct-2507 | 62.50 | 96.74 | 50.12 | 69.79 |

trajectory signals as the discriminative source for targeting. Below, we analyze why the anchor/non-anchor token split policy is more crucial.

**Static-confidence masking as a controlled counterexample.** To directly contrast trajectory-aware targeting with confidence-based heuristics, we implement two baselines that mask the same token fraction as T3S (about 20%): masking the top-$p$ highest-confidence tokens and masking the top-$p$ lowest-confidence tokens under the base model. Despite matching the mask budget, both baselines perform far worse than T3S and still exhibit Imitation Shock during training (Appendix K). This establishes that the effect is not driven by the mask budget or by masking tokens that are merely "easy" or "hard" at initialization.

**Functional validity of the T3S split via loss-transfer asymmetries.** T3S identifies a **functionally meaningful** partition: anchors and non-anchors exhibit strong within-group generalization but strongly antagonistic cross-group transfer. We verify this with a controlled loss-transfer experiment that trains on one token subset at a time and measures loss changes on all subsets; the resulting transfer matrix reveals a clear asymmetric suppression pattern consistent with anchor dominance (Table 6). Together, these results support

the key takeaway: **training-trajectory-aware token profiling is necessary** to expose and suppress early-dominating tokens and to clear the optimization path for the remaining yet-to-learn tokens.

## 5. Conclusion

We show that continual reasoning distillation exhibits **Imitation Shock**, where performance drops to an almost fixed **Imitation Bottleneck** and then recovers even as training loss decreases, and pre-bottleneck updates can be unnecessary or even harmful. We explain this behavior via token-level dynamics: token confidence splits into steadily increasing **Imitation-Anchor Tokens** and suppressed yet-to-learn tokens, where anchor learning delays progress on the latter. And the characteristic that these two types of tokens cannot coexist is the root cause of the failure in continual distillation. Based on this trajectory-defined mechanism, we propose **Training-Trajectory-Aware Token Selection (T3S)** to prevent early anchors from dominating optimization and to prioritize **yet-to-learn** tokens, yielding consistent gains in both AR and dLLM settings.

## Acknowledgements

This work is supported by the Fundamental and Interdisciplinary Disciplines Breakthrough Plan of the Ministry of Education of China. This paper is also supported by the National Regional Innovationand Development Joint Fund (No. U24A20254) and Ant Group Research Intern Program.

## Impact Statement

This paper presents work whose goal is to advance the field of Machine Learning. Specifically, we introduce Training-Trajectory-Aware Token Selection (T3S), a framework that uncovers and mitigates the "Imitation Shock" phenomenon in reasoning distillation. By dynamically masking easy "anchor" tokens to prevent the suppression of complex reasoning paths, our approach significantly enhances the stability and data efficiency of student model training. The potential societal consequences of our work are similar to those of existing language models, and standard safety and responsible deployment practices remain applicable.

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

## A. Additional Experiment Results

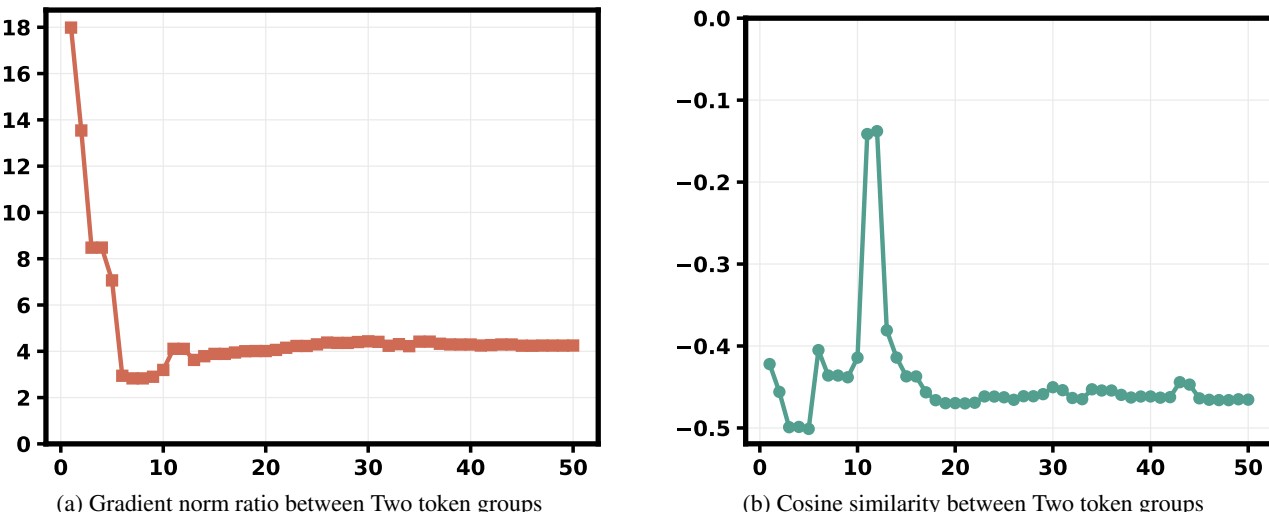

(a) Gradient norm ratio between Two token groups

(b) Cosine similarity between Two token groups

*Figure 8.* **Gradient dynamics between Anchor and Other token groups.** Panel (a) reports the ratio between Anchor-token and Other-token gradient norms across checkpoints. Panel (b) shows the cosine similarity between the two group gradients. Early in training, Anchor gradients dominate in magnitude and strongly conflict with Other-token gradients, supporting the view that anchor learning suppresses the optimization of yet-to-learn tokens.

*Table 6.* **Loss-transfer matrix between token subsets.** Anchor/Reasoning are split into Easy/Hard halves by base-model confidence within each group. Entries report % loss change on the evaluated subset after training on the row subset (negative is better). Within-group learning generalizes (Anchor ↔ Anchor, Reasoning ↔ Reasoning), as training on one half reduces loss on the other half. Cross-group transfer, however, is largely antagonistic. Details can be seen in Appendix L.

| Train on ↓ / Eval on → | Anchor-Easy | Anchor-Hard | Reasoning-Easy | Reasoning-Hard |
|---|---|---|---|---|
| **Anchor-Easy** | $-26.46$ | $-22.06$ | $+22.74$ | $+513.12$ |
| **Anchor-Hard** | $-17.17$ | $-37.14$ | $+30.70$ | $+165.90$ |
| **Reasoning-Easy** | $+57.37$ | $+74.59$ | $-17.11$ | $-82.84$ |
| **Reasoning-Hard** | $+34.83$ | $+12.48$ | $-4.87$ | $-94.58$ |

## B. Related Work

**Efficient Reasoning Distillation.** Early instruction-tuning studies already show that **high-quality subset selection** can yield substantially better performance than training on the full dataset: synthetic instruction corpora are often generated by strong LLMs and then filtered (Chen et al., 2023a; Wang et al., 2023; Xu et al., 2023; Shen et al., 2025a; Qin et al., 2025), and a series of data-selection methods consistently demonstrate that carefully chosen subsets outperform raw corpora (Tunstall et al., 2023; Zhang et al., 2024; Chen et al., 2023b; Zhao et al., 2024; Park, 2025). Building on this data-efficiency principle, LIMA (Zhou et al., 2023a) shows that about 1K curated samples suffice to induce strong behaviors, and subsequent works bring the same quality/difficulty-driven recipe into the **reasoning** domain, enabling strong CoT-style distillation with only hundreds to thousands of curated examples (Muennighoff et al., 2025; Ye et al., 2025b; Wen et al., 2025). In adjacent efficient distillation settings, recent work has focused on improving **who/what to distill** via diverse rollouts (Wu et al., 2025) or iterative model merge (Shen et al., 2025b), and **how to spend compute** via truncating long sequences (Chen et al., 2025). Nevertheless, these efforts largely remain **single-stage** in nature and do not address **continual distillation failure**, where iterative distillation can become unstable or collapse as the student approaches teacher-level reasoning. Our work fills this gap by diagnosing continual distillation failure from a token-level perspective.

**Token Priority Methods.** Another relevant direction involves methods that assign different importance to tokens or positions during training, in order to guide learning. In RL-based alignment, token-level credit assignment has become increasingly common: RTO (Zhong et al., 2025) formulates RLHF as a token-level MDP and learns token-wise rewards,

*Table 7.* **T3S-simple vs. T3S.** Fixed-step heuristic selector (step 10) remains strong.

| Dataset | Method | AIME24 | AIME25 |
|---------|--------|--------|--------|
| BOBA-200 | Base | 75.83 | 67.08 |
| | SFT (R1) | 71.25 | 55.00 |
| | T3S-simple (R1) | 78.83 | 70.00 |
| | T3S (R1) | **80.63** | **73.96** |
| S1K-200 | Base | 75.83 | 67.08 |
| | SFT (R1) | 72.50 | 55.83 |
| | T3S-simple (R1) | 77.50 | 70.21 |
| | T3S (R1) | **80.00** | **73.13** |

*Table 8.* **Selector checkpoint sweep.** AIME24/25 averaged score peaks at the true bottleneck, but remains above SFT for a broad range of selectors.

| Selector checkpoint | AIME24/25 Avg |
|---------------------|---------------|
| Base | 71.46 |
| SFT | 63.13 |
| T3S-ckt 2 | 68.34 |
| T3S-ckt 4 | 72.50 |
| T3S-ckt 6 | 74.52 |
| T3S-ckt 8 (true bottleneck) | **77.30** |
| T3S-ckt 10 | 74.42 |
| T3S-ckt 20 | 73.17 |
| T3S-ckt 30 | 71.25 |
| T3S-ckt 40 | 70.11 |
| T3S-ckt 50 | 69.48 |

while recent RLVR/RLFT analyses and objectives further highlight that only a small set of "decision" tokens matter most—e.g., optimizing only high-entropy forking tokens can match full-gradient RLVR (Wang et al., 2025b), and encouraging exploration specifically on critical tokens improves RL fine-tuning efficiency (Vassoyan et al., 2025); closely related, Lin et al. (2024b) identify *critical tokens* in reasoning trajectories and leverage token-level contrastive estimation to improve DPO-style training. In continual pretraining, RHO-1 (Lin et al., 2024a) proposes Selective Language Modeling that filters easy/uninformative tokens, improving data efficiency. In length-/token-budget curricula for reasoning, Hammoud et al. (2025) start with generous budgets and gradually tighten them to encourage exploration then compression, yielding better token-efficiency. Together, these lines of work (Shen et al., 2026) not only highlight the importance of token selection, but also suggest that *priority rules are inherently setting-dependent*: what constitutes "important" tokens differs across RL alignment, pretraining, and budgeted reasoning. Motivated by this perspective, we design a token-priority mechanism tailored to the continual distillation setting and further show that token-entropy-based criteria (effective in some RLVR analyses) do not directly transfer to our regime in Section 4.5.

**Diffusion Language Models.** Diffusion LLMs (dLLMs/MDLMs) reformulate text generation as iterative denoising with random masking and bidirectional conditioning, evolving from early discrete corruption to scalable architectures and simplified objectives (Li et al., 2022; He et al., 2023; Lou et al., 2023; Sahoo et al., 2024; Song et al., 2025). To reduce cost and narrow the gap to AR models, recent work explores training from scratch (e.g., LLaDA and MoE variants) as well as AR-initialized diffusion via attention-mask annealing or direct conversion (Nie et al., 2025; Zhu et al., 2025; Ye et al., 2025a). Block Diffusion LMs further hybridize the two paradigms by defining an autoregressive distribution **over blocks** while denoising tokens **within each block** via diffusion (Arriola et al., 2025), a structure we view as crucial for transferring the cross-model token signals exploited by T3S. Finally, diffusion-style training is often a "super data learner" that trades higher data utilization for larger compute budgets (Ni et al., 2025); under the inherently redundant random-mask dynamics, this amplifies the importance of token selection/priority, yet it remains relatively under-explored, motivating our extension of T3S to the dLLM setting.

*Table 9.* **Cross-model selection of yet-to-learn tokens (S1K-200, DeepSeek-R1 distillation).** Cross-model selection uses the token mask produced by the other model in the same Qwen3 family (shared tokenizer).

| Training model | Training method | AIME24 | AIME25 | AVG |
|---|---|---|---|---|
| Qwen3-8B | Base | 75.83 | 67.08 | 71.46 |
| | SFT | 72.50 | 55.83 | 64.17 |
| | T3S (self-select) | 80.00 | 73.13 | 76.57 |
| | T3S (cross-model select) | 77.50 | 70.00 | 73.75 |
| Qwen3-32B | Base | 81.46 | 72.08 | 76.77 |
| | SFT | 76.67 | 67.50 | 72.09 |
| | T3S (self-select) | 83.33 | 73.33 | 78.33 |
| | T3S (cross-model select) | 83.33 | 72.50 | 77.92 |

## C. Cross-Model Transferability of Yet-to-Learn Tokens

We further test whether the *yet-to-learn* tokens identified by T3S are purely model-specific, or whether they exhibit *cross-model consistency* within a tokenizer-sharing model family. Concretely, under the S1K-200 distillation setting with DeepSeek-R1 as the teacher, we swap the token selector across Qwen3-8B and Qwen3-32B: we use the yet-to-learn token mask computed by Qwen3-8B to train Qwen3-32B, and vice versa. We compare against standard SFT and the default T3S setting where each model uses its own selector.

These results indicate that, while T3S is nominally model-aware, the yet-to-learn token sets are to a large extent *transferable* across models that share the same tokenizer. Cross-model selection is consistently weaker than self-selection, but remains far stronger than standard SFT, suggesting that the dominant barriers captured by yet-to-learn token targeting are partially shared within a model family. Due to resource constraints we do not scale this experiment further, but the observed transferability motivates a promising direction: using a small (few-billion-parameter) selector to identify yet-to-learn tokens for a much larger (hundreds-of-billions or trillion-scale) tokenizer-sharing flagship model.

## D. T3S for dLLMs

### D.1. Preliminary

**Diffusion-style language models (dLLMs).** A dLLM is trained to reconstruct masked tokens given visible tokens. Let $m \in \{0,1\}^T$ be a binary mask where $m_t = 1$ denotes "masked". We write $y_{\neg m}$ for visible tokens and $y_m$ for masked tokens. Training samples a mask $m \sim \mathcal{M}$ and minimizes the conditional negative log-likelihood:

$$\mathcal{L}(\phi) = \mathbb{E}_{x,y} \, \mathbb{E}_{m \sim \mathcal{M}} \left[ \sum_{t:m_t=1} -\log p_\phi(y_t \mid y_{\neg m}, x, m) \right]. \tag{6}$$

### D.2. T3S for dLLMs: Targeting Bottleneck-Candidate Tokens via Union Masking

Unlike AR models, dLLMs are trained with an intrinsic random-masking objective, learning by reconstructing masked positions under partial observations. Thus, the key question is not which tokens to imitate most aggressively, but which tokens should be **preferentially masked** so the model is forced to learn what it still does not know. Accordingly, we aim to encourage dLLMs to repeatedly practice the yet-to-learn tokens under **any set of already generated (visible) tokens**, rather than wasting capacity on redundant or already-easy tokens sampled by uniform random masking. We further find that **yet-to-learn** tokens exhibit non-trivial cross-model consistency within tokenizer-sharing model families (Appendix C), suggesting these yet-to-learn tokens exhibit a degree of cross-model consistency within model families that share a tokenizer. Since dLLMs do not provide a clean AR-style token-wise likelihood interface for precise token-level localization, we use an AR selector as a practical proxy to identify yet-to-learn tokens for dLLM training, especially for block-diffusion training (Arriola et al., 2025), which defines an autoregressive probability distribution over blocks of discrete random variables.

Formally, let $\mathcal{B}(x,y) \subseteq [T]$ denote the targeted (yet-to-learn) positions. We would like to minimize the expected reconstruc-

tion loss on $\mathcal{B}(x, y)$ under a generic visibility pattern $y_{\neg m}$:

$$\min_{\phi} \ \mathbb{E}_{x,y} \ \mathbb{E}_{m \sim \mathcal{M}} \left[ \sum_{t \in \mathcal{B}(x,y)} - \log p_{\phi}(y_t \mid y_{\neg m}, x, m) \right], \tag{7}$$

which captures the requirement that the model should be able to produce yet-to-learn tokens under arbitrary partial observations induced by $m$. Trajectory-aware targeting provides a principled way to instantiate $\mathcal{B}(x, y)$ and bias training toward these tokens, helping dLLMs escape the heavy, low-yield randomness of pure random masking.

**Yet-to-learn-token set.** We therefore define a set of **yet-to-learn** tokens as those whose confidence drops significantly by the Imitation Bottleneck:

$$\mathcal{B}(x, y) = \{t \in [T] \mid \Delta c_t(x, y) < -\tau\}, \tag{8}$$

where $\tau > 0$ controls the severity. $\mathcal{B}(x, y)$ is computed offline using the AR selector $M_0$. Operationally, this rule extracts positions that become substantially less predictable during Imitation Shock; empirically, these positions concentrate on yet-to-learn tokens, making them an effective target set for dLLM training.

**Union masking for dLLM training.** Given a random mask $m \sim \mathcal{M}$, we augment it by forcing all targeted tokens to be masked:

$$m' = m \ \vee \ \mathbf{1}_{\mathcal{B}(x,y)}, \tag{9}$$

where $\vee$ is element-wise OR and $\mathbf{1}_{\mathcal{B}}$ is the indicator mask of $\mathcal{B}(x, y)$. We then train the dLLM with $m'$:

$$\mathcal{L}_{\text{T3S-dLLM}}(\phi) = \mathbb{E}_{x,y} \ \mathbb{E}_{m \sim \mathcal{M}}$$
$$\left[ \sum_{t: m'_t = 1} - \log p_{\phi}(y_t \mid y_{\neg m'}, x, m') \right]. \tag{10}$$

This enforces that the dLLM repeatedly practices generating $\mathcal{B}(x, y)$ across diverse visible-token conditions (from random $m$), directly strengthening its ability to produce yet-to-learn tokens under arbitrary partial observations. As a secondary interpretation, the difficulty of reliably generating these yet-to-learn tokens may itself be the latent bottleneck that limits dLLM reasoning quality.

### D.3. Main Experiments

**Experimental setting.** We evaluate T3S in the diffusion-style language modeling (dLLM) setting. We distill on the full S1K dataset; however, due to the block-diffusion attention mechanism and the limited effective context length of current dLLM architectures, training is restricted to at most 16K tokens of usable context. We therefore select **Qwen3-235B-Instruct-2507** as the teacher model, whose reasoning traces are comparatively shorter while maintaining high reasoning quality.

As the student model, we use **LLaDA-2.0-Mini** (Bie et al., 2025) as the base dLLM. We train with a learning rate of $1 \times 10^{-5}$, batch size 64, and 20 training epochs. We apply T3S to reconstruct the distillation dataset at the token level, following the same training trajectory-aware token selection procedure described earlier. For comparison, we include the base model, standard SFT, and a set of strong autoregressive baselines.

**Results.** Table 5 shows that T3S yields substantial gains in the dLLM setting. When trained on the T3S-reconstructed dataset, LLaDA-2.0-Mini improves dramatically over both the base model and standard SFT, with particularly large gains on AIME25. On average, the T3S-trained dLLM outperforms its autoregressive counterpart (Ling-Mini) despite sharing the same architecture, and surpasses all other compared frontier models in the 16B parameter regime under non-thinking settings. These results demonstrate that the benefits of training trajectory-aware token selection are not limited to autoregressive models. Even under the more challenging block-diffusion training paradigm, explicitly targeting trajectory-identified tokens enables the model to acquire substantially stronger reasoning capability.

*Table 10.* **Total generated tokens under different training strategies.** Lower is better.

| Model | AIME25 | MATH500 | TheoremQA |
|---|---|---|---|
| Base | 5,685,312 | 4,373,220 | 2,280,907 |
| SFT | 5,248,832 | 4,537,956 | 2,258,411 |
| T3S | **4,675,392** | **4,210,788** | **1,699,091** |

*Table 11.* **Sensitivity of T3S-dLLM to $\tau$.** We report the fraction of selected tokens $|\mathcal{B}(x,y)|/T$ and AIME25 score.

| $\tau$ | Selected tokens (%) | AIME25 |
|---|---|---|
| 0.0 | 81.36 | 32.92 |
| 0.1 | 10.36 | 40.00 |
| **0.2** | **5.24** | **53.33** |
| 0.3 | 2.52 | 46.67 |
| 0.4 | 1.13 | 43.33 |

### D.4. Inference efficiency.

Beyond accuracy improvements, T3S also leads to more efficient inference. We measure the total number of generated tokens on evaluation benchmarks and report the results below. T3S reduces the total number of generated tokens by approximately 15–25% across benchmarks compared to the base and SFT models. We hypothesize that once a dLLM is successfully trained to reliably generate key reasoning tokens, these tokens may act as more accurate and efficient internal "anchors" for inference, guiding the model toward shorter and more direct reasoning trajectories. This suggests that improving token-level learning not only enhances reasoning accuracy but also implicitly shapes more efficient inference paths in diffusion-style models.

### D.5. Sensitivity of T3S-dLLM to the Threshold $\tau$

T3S for dLLMs targets bottleneck-candidate tokens $\mathcal{B}(x,y) = \{t \in [T] \mid \Delta c_t(x,y) < -\tau\}$ (Eq. (8)) and forces their inclusion in the training mask via union masking (Eq. (9)). The threshold $\tau$ controls the size of $\mathcal{B}(x,y)$ and thus mediates a key trade-off: if $\tau$ is too small, the targeted set becomes overly large, weakening the intrinsic random-mask nature of dLLM training and effectively pushing the objective toward a nearly deterministic "generate almost everything" regime; if $\tau$ is too large, the targeted set becomes too sparse and T3S gradually collapses toward the pure random-masking baseline. We therefore choose $\tau = 0.2$, which selects roughly the top ∼5% most severely dropped tokens and preserves stochastic masking while injecting a focused supervision signal. Table 11 confirms this choice: performance peaks near $\tau = 0.2$, while $\tau = 0$ (81.36% tokens selected) degrades performance sharply and $\tau \geq 0.3$ reduces gains as the target set shrinks.

## E. Universality of Imitation Shock: Additional Settings

We further study the universality of *Imitation Shock* under a broader set of conditions, covering different teacher models, datasets, dataset scales, student models, and domains. Unless otherwise noted, we follow the same distillation hyperparameters as in the main experiments and report checkpoint-wise trajectories. For **math reasoning** settings, we use **AIME25** (Math-AI, 2025) as the primary performance metric; for the **code reasoning** setting, we use **LiveCodeBench** (Jain et al., 2024).

**(1) Different teacher model.** We distill Qwen3-8B from **QWQ**, a widely regarded in-distribution teacher for small Qwen-series models (Shen et al., 2025b; Wu et al., 2025), using the same BOBA-200 prompts and evaluation protocol. We observe the same performance crash-and-recovery pattern (Figure 2(a)).

**(2) Different dataset.** We repeat DeepSeek-R1→Qwen3-8B distillation on **S1K-200** (Muennighoff et al., 2025) (math reasoning) with the same training configuration. Imitation Shock persists (Figure 2(b)), indicating the phenomenon is not specific to a single dataset.

**(3) Larger dataset scale.** To go beyond the efficient-distillation regime, we distill Qwen3-8B from DeepSeek-R1 on **OpenThought3** (Guha et al., 2025), which has ~65K samples. Even though training does not cover a full epoch, we still observe a clear Imitation Bottleneck as the lowest valley point (Figure 2(c)). Compared to the small-data setting, post-bottleneck performance can exhibit larger fluctuations, suggesting dataset-specific variations, while the teacher-induced surface characteristics may be shared across datasets.

**(4) Different student model.** We further test whether the phenomenon depends on the Qwen3-8B backbone by distilling **Qwen3-235B** on BOBA-200 from a different teacher family, **R1-distill Llama3-8B** (deepseek ai, 2025). Imitation Shock remains (Figure 2(d)), indicating the effect is not tied to a specific student series.

**(5) Different domain.** Beyond math reasoning, we sample a code subset of comparable scale from **RL-Code-Math-v5** (typhoon ai, 2025) and distill Qwen3-8B from **QWQ**. Using **LiveCodeBench** as the metric, we again observe the same crash-and-recovery dynamics (Figure 2(e)), suggesting that Imitation Shock is not confined to math reasoning but extends to code reasoning domains as well.

# F. Additional Distillation Dynamics

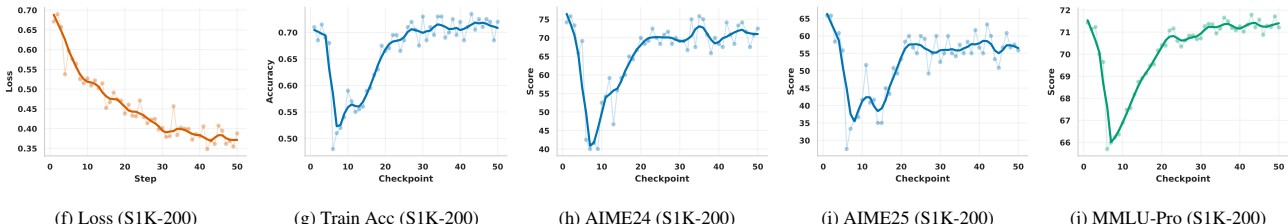

| (f) Loss (S1K-200) | (g) Train Acc (S1K-200) | (h) AIME24 (S1K-200) | (i) AIME25 (S1K-200) | (j) MMLU-Pro (S1K-200) |

*Figure 9.* **Imitation Shock under DeepSeek-R1 distillation on S1K-200.** The same sharp performance drop and recovery pattern observed on BOBA-200 also appears on S1K-200.

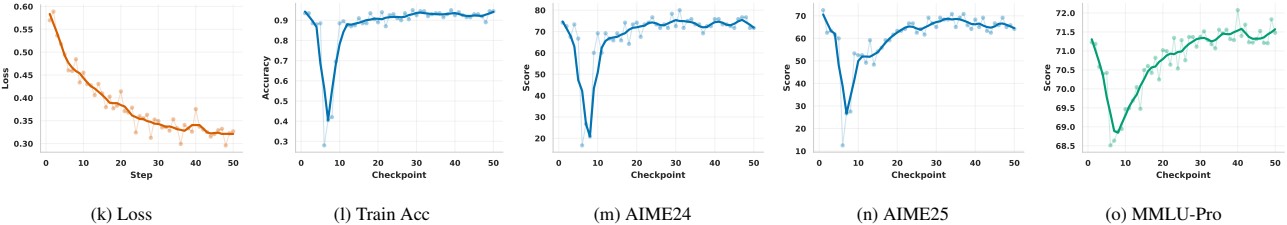

| (k) Loss | (l) Train Acc | (m) AIME24 | (n) AIME25 | (o) MMLU-Pro |

*Figure 10.* **Imitation Shock persists under an in-distribution teacher (QWQ) on BOBA-200.**

# G. OOD Reasoning Gains and Forgetting Mitigation

To assess whether T3S improves general reasoning beyond AIME-style evaluation and whether it mitigates catastrophic forgetting, we evaluate distilled students on two task families:

**Reasoning-related tasks.** We report performance on PhyBench (Meng et al., 2024), GPQA-Diamond (GPQA-D) (Rein et al., 2023), LiveCodeBench (LCB) (Jain et al., 2024), and MMLU-Pro (Wang et al., 2024), and compute **Delta_Avg** as the average change relative to the Base model across these reasoning benchmarks.

**Forgetting-related tasks.** To quantify retention of instruction-following and general knowledge behavior, we evaluate SuperGPQA (Du et al., 2025), CEval (Seifert et al., 2024), and IFEval (Zhou et al., 2023b), and compute **Delta_Avg** as the average change relative to the Base model across these retention benchmarks.

Table 12 and Table 13 summarize results under both teachers (DeepSeek-R1 and QWQ) and both distillation datasets (BOBA-200 and S1K-200). Overall, T3S consistently improves *OOD reasoning* over SFT across nearly all settings, with particularly clear gains on PhyBench and LiveCodeBench, indicating better transfer to harder or more distribution-shifted reasoning

*Table 12.* **OOD reasoning gains of T3S.** We evaluate on PHYBENCH, GPQA-D, LCB, and MMLU-Pro. ΔAvg is computed as the average score change relative to the Base model across these reasoning benchmarks (higher is better).

| Dataset | Teacher (Method) | PHYBENCH | GPQA-D | LCB | MMLU-Pro | ΔAvg |
|---|---|---|---|---|---|---|
| Base | | 20.47 | 57.77 | 55.76 | 71.42 | – |
| BOBA-200 | R1 (SFT) | 21.79 | 56.34 | 53.41 | 69.92 | ↓-0.99 |
| | R1 (T3S) | **23.95** | **59.47** | **59.32** | **72.65** | ↑2.49 |
| | QWQ (SFT) | 19.55 | **58.33** | 56.14 | 71.48 | ↑0.02 |
| | QWQ (T3S) | **23.76** | 58.21 | **58.68** | **72.88** | ↑2.03 |
| S1K-200 | R1 (SFT) | 20.24 | 58.33 | 55.99 | 71.21 | ↑0.09 |
| | R1 (T3S) | **26.26** | **58.59** | **59.58** | **72.65** | ↑2.92 |
| | QWQ (SFT) | 22.18 | 57.32 | 57.63 | 71.73 | ↑0.86 |
| | QWQ (T3S) | **24.36** | **58.59** | **58.83** | **73.09** | ↑2.36 |

*Table 13.* **Forgetting mitigation of T3S.** We evaluate on SUPER-GPQA, CEval, and IFEval. ΔAvg is computed as the average score change relative to the Base model across these retention benchmarks (higher is better).

| Dataset | Teacher (Method) | SUPER-GPQA | CEval | IFEval | ΔAvg |
|---|---|---|---|---|---|
| Base | | 10.51 | 83.58 | 83.60 | – |
| BOBA-200 | R1 (SFT) | 10.12 | 82.84 | 82.49 | ↓-0.75 |
| | R1 (T3S) | **10.29** | **84.40** | **83.20** | ↑0.07 |
| | QWQ (SFT) | 10.21 | 82.17 | 81.52 | ↓-1.26 |
| | QWQ (T3S) | **10.40** | **83.58** | **83.83** | ↑0.04 |
| S1K-200 | R1 (SFT) | 10.21 | 82.88 | 83.23 | ↓-0.46 |
| | R1 (T3S) | **10.33** | **83.80** | **83.55** | ↑0.00 |
| | QWQ (SFT) | 10.29 | 83.12 | 82.53 | ↓-0.58 |
| | QWQ (T3S) | **10.54** | **83.51** | **84.84** | ↑0.40 |

workloads. At the same time, T3S yields *smaller degradation* (or even slight improvements) on the forgetting-related suite compared to SFT, suggesting that suppressing anchor-token dominance helps avoid overwriting general instruction-following and knowledge behaviors during continual distillation.

# H. Additional Experiments: Scale, Baselines, Model Families, and Domains

## H.1. Scaling to Larger Distillation Corpora

We further evaluate T3S beyond the 200-sample efficient-distillation regime by scaling the distillation data significantly. Table 14 shows that under an out-of-distribution teacher (DeepSeek-R1), direct continual distillation can still degrade severely at the 3K scale, while T3S prevents collapse and yields strong net gains. Moreover, under a massive in-distribution setting (BOBA ∼100K with Qwen3-235B as teacher), T3S continues to amplify the gains beyond standard SFT.

## H.2. Comparison with Existing Distillation Strategies

We compare T3S against representative distillation strategies that also aim to address early-stage mismatch. A key practical limitation of some baselines is their reliance on teacher-side token-level signals and/or strict tokenizer sharing, which breaks our core OOD-teacher setting (e.g., DeepSeek-R1 → Qwen3). Nevertheless, under a tokenizer-sharing setting (Teacher: Qwen3-32B, Student: Qwen3-8B), T3S still outperforms both GKD (Agarwal et al., 2023) and SWITCH (Koo et al., 2024)(Table 15). We further compare against Mix Distillation (Li et al., 2025) under two teacher pairings and find T3S consistently yields higher AIME24/25 averages (Table 16).

*Table 14.* **Scaling up distillation data.** T3S remains effective from 3K to ∼100K scale.

| Dataset (Scale) | Method | AIME24 | AIME25 |
|---|---|---|---|
| OpenThought3-3K | Base | 75.83 | 67.08 |
| | SFT | 60.00 | 45.00 |
| | T3S | **77.50** | **68.64** |
| BOBA-100K | Base | 75.83 | 67.08 |
| | SFT | 75.83 | 70.00 |
| | T3S | **80.21** | **72.71** |

*Table 15.* **Comparison with GKD and SWITCH (tokenizer-sharing setting).**

| Method | AIME24 | AIME25 |
|---|---|---|
| GKD (Agarwal et al., 2023) | 73.33 | 53.33 |
| SWITCH (Koo et al., 2024) | 75.83 | 66.67 |
| T3S (Ours) | **80.83** | **70.00** |

### H.3. Generality Across Model Families

To test whether T3S depends on a specific student series, we apply it to a different student architecture by continually distilling Qwen3-235B from R1-Distill-Llama3-8B (Wang et al., 2025c). T3S remains effective and substantially outperforms direct SFT (Table 17).

### H.4. Additional Domain: Code Reasoning

Beyond math reasoning, we also observe Imitation Shock on a purely code-based dataset and apply T3S under the same continual distillation setup. On LiveCodeBench, T3S improves over SFT and the base model (Table 17).

## I. Experimental Setup Details

**Datasets.** We use two high-quality open-source mathematical reasoning datasets: BOBA (inclusionAI, 2025) and S1K (Muennighoff et al., 2025). Following previous work (Shen et al., 2025b), from each dataset, we sample 200 prompts and denote the resulting subsets as **BOBA-200** and **S1K-200**. For BOBA-200, we directly use the default 200 problems released with the BOBA benchmark. For S1K-200, we remove proof-style problems without verifiable final answers and uniformly sample 200 remaining prompts with boxed answers. The same set of prompts is used across all compared methods and teachers to avoid cherry-picking and reduce sensitivity to particular samples.

**Teachers and distillation data.** For each prompt, we query two representative teacher models that exhibit contrasting distributional relationships with Qwen3-8B: **DeepSeek-R1** (Guo et al., 2025), a canonical out-of-distribution teacher, and **QWQ** (Team, 2024), a widely regarded best teacher for small Qwen-series models (Shen et al., 2025b; Wu et al., 2025). Each teacher generates 16 reasoning traces with temperature 0.6 and a maximum length of 32,768 tokens. For distillation, we randomly select one correct reasoning trace as the training target; if none of the generated traces is correct, the corresponding prompt is discarded for that teacher's distillation corpus.

**Training configuration and evaluation.** All experiments fine-tune **Qwen3-8B** (Yang et al., 2025) as the student model. Unless otherwise specified, we use a batch size of 64, a learning rate of $1 \times 10^{-5}$, and train for 50 optimization steps. For standard SFT baselines, we train on the corresponding distilled corpus and report the final checkpoint. For the Recovering Residual Transfer (RRT) method, we identify the **Imitation Bottleneck** based on training accuracy, construct recovering residual using post-bottleneck updates only, and apply them to the base model. For T3S and its ablation variants, we follow the same training configuration and modify only the token-level masking strategy. All evaluations are conducted using AIME24 (Math-AI, 2024) and AIME25 (Math-AI, 2025), with each score reported as the average over 16 runs.

*Table 16.* **Comparison with Mix Distillation ([Li et al., 2025](#)).** Reported as AIME24/25 average.

| Teachers (large/small) | Mix Distillation | T3S |
|---|---|---|
| R1 / QWQ | 68.75 | **75.84** |
| Qwen3-235B / QWQ | 71.67 | **77.40** |

*Table 17.* **Cross-family distillation.** Student: Qwen3-235B; Teacher: R1-Distill-Llama3-8B.

| Method | AIME24 | AIME25 |
|---|---|---|
| Base | 45.00 | 33.33 |
| SFT | 50.00 | 36.67 |
| T3S | **56.04** | **55.83** |

# J. Why Static Signals are Insufficient for Targeting

This appendix section studies whether the anchor/non-anchor partition discovered by T3S can be recovered from *static* signals available at a single checkpoint, without using training-trajectory information. Such alternatives are closely related to recent directions that select tokens by initial uncertainty (e.g., token entropy / base-model confidence) or by local gradient geometry. Across both analyses below, we find a consistent conclusion: **static signals do not yield a robust separation**. This supports a core design choice of T3S: **trajectory-based confidence changes** provide a uniquely informative signal for identifying the tokens that dominate early optimization in continual reasoning distillation.

## J.1. Why Gradient Similarity is Insufficient: Token-Gradient Sketch Visualization

To further test whether the anchor/non-anchor partition can be recovered from *instantaneous* gradient information, we perform a token-level gradient-sketch analysis on the same checkpoint. Intuitively, if anchor tokens and reasoning-execution tokens were separable by local gradient geometry, one might hope to identify anchors via gradient similarity alone, without using trajectory signals. However, we find that **token gradients do not yield a clean separation**: anchor and non-anchor tokens substantially overlap in low-dimensional projections, suggesting that *trajectory-based confidence changes* are necessary for reliable targeting.

**Token-gradient sketches.** We follow a randomized JVP-based sketching procedure to approximate per-token gradients with respect to the LoRA parameters. Concretely, for each token position, we compute a length-$k$ sketch vector by projecting the per-token loss gradient onto $k$ random directions (via Jacobian–vector products). We then visualize all token sketches in 2D, coloring tokens by the sign of their trajectory-based confidence change $\Delta c_t$ (anchor if $\Delta c_t > 0$, otherwise non-anchor). Together with the loss-transfer asymmetries (Table [6](#)), this result supports a key design choice of T3S: **only Training-Trajectory-Aware token profiling reliably exposes the specific tokens that dominate early optimization**, whereas gradient-only heuristics fail to provide a robust separating signal.

## J.2. Why Initial Confidence is Insufficient for Targeting

A natural alternative to trajectory-aware targeting is to partition tokens using their *initial* confidence under the base model (e.g., treating low-confidence / high-loss tokens as the primary "to-be-learned" set). If imitation anchors were simply the tokens the base model is initially uncertain about, then such a heuristic could identify anchors without tracking the distillation trajectory. However, we find that **the anchor/non-anchor split cannot be explained by initial confidence alone**.

**Anchor tokens are not just "low-confidence" tokens.** Figure [12](#) visualizes the base-model confidence distribution for the two token groups (colored by the sign of trajectory-based confidence change $\Delta c_t$). Anchor tokens are *not* confined to the low-confidence region, and non-anchor tokens are *not* uniformly high-confidence. Instead, tokens across a broad range of base-model confidence exhibit different learning behaviors, indicating that the early dominance of anchors is **not** a trivial consequence of initialization difficulty.

**Implication.** Because initial-confidence heuristics mix the two groups, they cannot reliably prevent anchors from dominating early optimization. This motivates using *trajectory* signals (e.g., $\Delta c_t$ around the Imitation Bottleneck) for token

*Table 18.* Results on code-based dataset.

| Method | LiveCodeBench |
|--------|---------------|
| Base | 55.76 |
| SFT | 53.29 |
| T3S | 61.41 |

*Table 19.* **Initial-confidence masking baselines vs. T3S (BOBA-200, R1).** Highest-Top-$p$ and Lowest-Top-$p$ mask the same token fraction as T3S (about 20%), selected solely by base-model confidence.

| Method | AIME24 | AIME25 | AVG |
|--------|--------|--------|-----|
| BASE | 75.83 | 67.08 | 71.46 |
| SFT | 71.25 | 55.00 | 63.13 |
| Highest-Top-$p$ | 67.50 | 59.17 | 63.34 |
| Lowest-Top-$p$ | 71.67 | 65.00 | 68.34 |
| **T3S** | **80.63** | **73.96** | **77.30** |

targeting: **the evolution of confidence provides a discriminative signal that initialization-only criteria do not.**

**Takeaway.** Across both gradient-sketch visualization (Section J.1) and initial-confidence overlap (Section J.2), we find that **static, single-checkpoint signals do not reliably expose the anchor/non-anchor partition**. In continual reasoning distillation, the relevant tokens are characterized by *how their confidence evolves* along training—particularly around the Imitation Bottleneck. This motivates using training-trajectory-aware profiling as in T3S to identify and suppress early-dominating anchors and to clear optimization capacity for the remaining yet-to-learn tokens.

## K. Why Initial-Confidence Masking Fails and Does Not Remove Imitation Shock

We test whether token masking based solely on initial confidence can match T3S. Concretely, we construct two baselines that mask the same fraction of tokens as T3S (about 20%): (i) masking the top-$p$ *highest-confidence* tokens under the base model, and (ii) masking the top-$p$ *lowest-confidence* tokens under the base model. Despite matching the mask budget, both baselines perform far worse than T3S (Table 19). More importantly, both training runs still exhibit the characteristic *crash then recover* behavior of Imitation Shock (Figure 13). This indicates that the failure mode underlying continual distillation, and the effectiveness of T3S, are not tied to the initial-confidence distribution or to masking "easy" versus "hard" tokens in aggregate. Instead, they depend on a trajectory-defined subset of tokens whose early acquisition dominates optimization in a way that static confidence criteria cannot reliably identify.

## L. Loss-Transfer Analysis: Asymmetric Token Interference

This appendix provides a detailed description of the *loss-transfer matrix* experiment used to probe token-level interference and to validate that the token partition identified by T3S reflects a functional separation rather than a confidence-based artifact. The core question is: **if we update the model using gradients from only one token subset, which other token subsets benefit or get harmed?**

### L.1. Experimental setup

**Token groups from T3S.** We start from the T3S partition induced by trajectory-based confidence change $\Delta c_t$: tokens with $\Delta c_t > 0$ are treated as **Anchor** tokens, and tokens with $\Delta c_t \leq 0$ as **Other** (non-anchor) tokens. To control for initialization difficulty, we further split tokens *within* each group by their **base-model confidence** (equivalently, base loss), into two equal-sized halves:

$$\text{\textbf{Anchor-Easy}}, \text{ \textbf{Anchor-Hard}}, \text{ \textbf{Other-Easy}}, \text{ \textbf{Other-Hard}},$$

where "Hard" denotes the bottom 50% by initial confidence (higher loss) *within that group*. (We use "Other" here to emphasize that this split is defined by the trajectory-based anchor/non-anchor partition, not by any semantic label.)

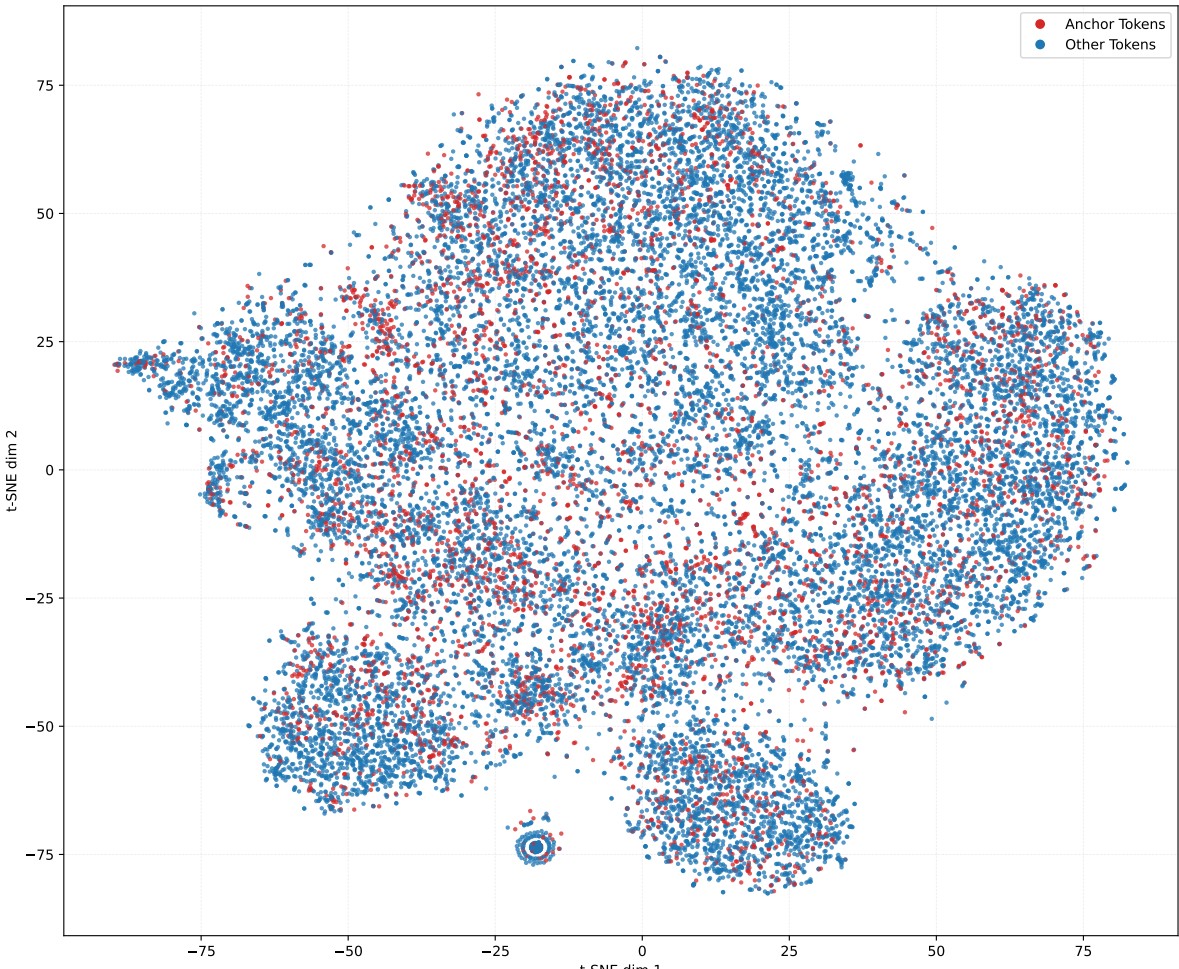

*Figure 11.* **Token-gradient sketch visualization does not cleanly separate anchors.** We project token-level LoRA gradient sketches into 2D (t-SNE/PCA-style embedding) and color tokens by $\operatorname{sign}(\Delta c_t)$ (anchor vs. non-anchor). The two groups heavily overlap, indicating that local gradient geometry alone is insufficient for reliably identifying imitation anchors.

**Subset-only training protocol.** Starting from the **same** base checkpoint $\theta_0$, we create four training runs. In each run, we optimize *only* one token subset by masking out all other token positions in the loss. Concretely, for a chosen subset $\mathcal{S} \subseteq [T]$, we minimize:

$$\mathcal{L}_{\mathcal{S}}(\theta) = \mathbb{E}_{(x,y)} \left[ \sum_{t \in \mathcal{S}} - \log p_{\theta}(y_t \mid y_{<t}, x) \right],$$

and set the weight of positions $t \notin \mathcal{S}$ to zero (i.e., they contribute no gradient). All four runs use the same data, optimizer, learning rate, batch size, and number of update steps as the main continual distillation setting, differing only in the subset mask.

**Evaluation: loss-transfer measurement.** After training on subset $\mathcal{S}$, we evaluate the **average per-token loss** on *each* subset $\mathcal{T} \in \{\text{Anchor-Easy, Anchor-Hard, Other-Easy, Other-Hard}\}$. Let $\ell_{\mathcal{T}}(\theta)$ denote the mean loss restricted to tokens in $\mathcal{T}$. We report the **percentage loss change**:

$$\Delta_{\mathcal{S} \to \mathcal{T}} = 100 \times \frac{\ell_{\mathcal{T}}(\theta_{\mathcal{S}}) - \ell_{\mathcal{T}}(\theta_0)}{\ell_{\mathcal{T}}(\theta_0)},$$

where $\theta_{\mathcal{S}}$ is the checkpoint obtained by training only on $\mathcal{S}$. Negative values indicate improvement (loss reduction), while positive values indicate harm (loss increase). Collecting $\Delta_{\mathcal{S} \to \mathcal{T}}$ for all pairs yields the $4 \times 4$ matrix in Table 6.

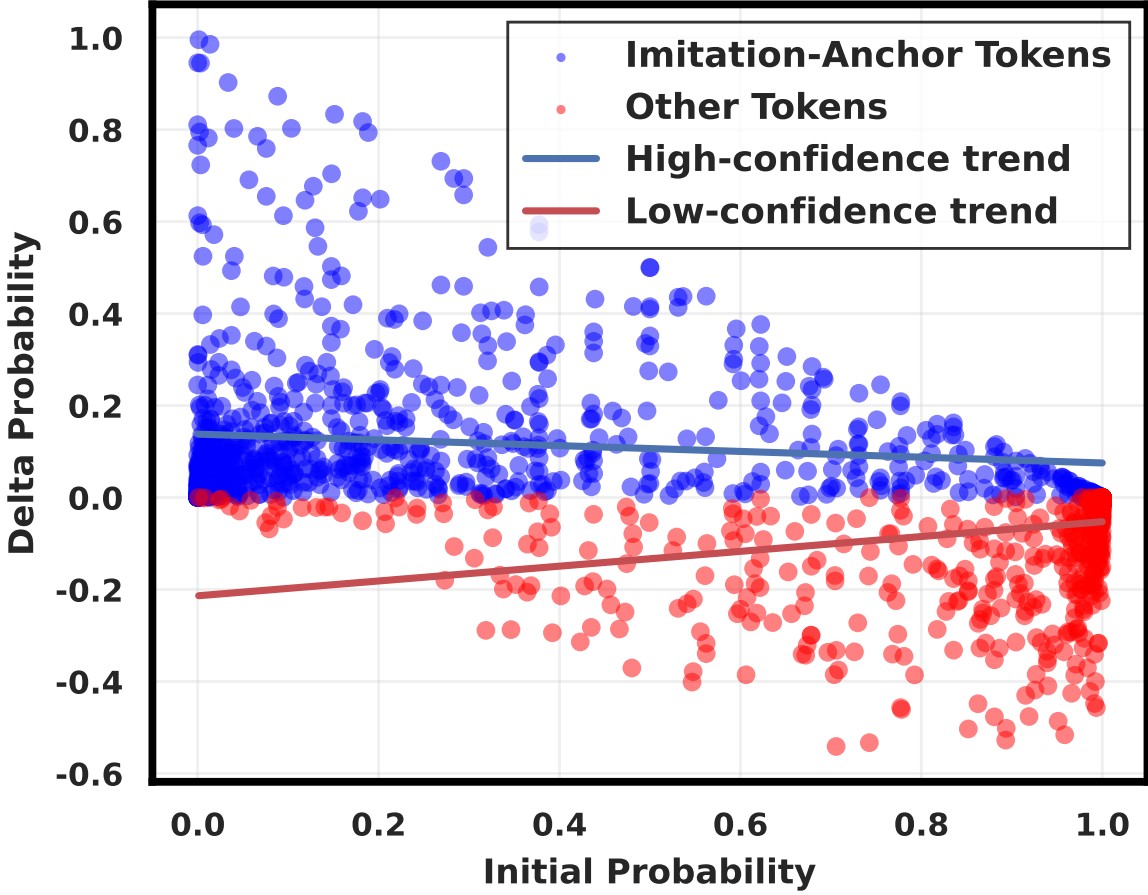

*Figure 12.* **Base-model confidence is insufficient to separate anchors.** We plot token groups using their base-model confidence and color by the sign of trajectory-based confidence change $\Delta c_t$ (anchor if $\Delta c_t > 0$, otherwise non-anchor). The two groups overlap substantially, showing that the anchor/non-anchor partition cannot be recovered by initial confidence alone.

### L.2. Results and interpretation

Table 6 reveals three robust patterns.

**(1) Within-group learning generalizes; cross-group transfer is antagonistic.** Training on Anchor-Easy or Anchor-Hard reduces loss on the other Anchor subset (e.g., Anchor-Easy $\rightarrow$ Anchor-Hard is negative), and similarly training on Other-Easy/Other-Hard reduces loss on the other Other subset. In contrast, cross-group transfer is broadly harmful: training on Anchors increases loss on Others, and training on Others increases loss on Anchors. This demonstrates that the T3S-derived partition corresponds to a **functional** split: gradients that help one group systematically hurt the other.

**(2) Why anchors dominate early: strong suppression in the Hard regime.** The antagonism is most striking for low-confidence (Hard) tokens. In particular, Anchor-Hard training substantially *increases* Other-Hard loss (Anchor-Hard $\rightarrow$ Other-Hard is strongly positive), while Other-Hard training decreases its own loss. This asymmetry implies that when optimization is driven by anchor gradients early on, it can actively block progress on the yet-to-learn tokens. This supports the causal storyline behind Imitation Shock: early training can be pulled onto an anchor-driven trajectory that suppresses the acquisition of the remaining tokens, causing broad performance collapse.

**(3) Why trajectories matter: anchors create a strong and stable learning signature.** Within-group "Hard$\rightarrow$Easy" transfer is stronger for anchors than for others, meaning anchor learning produces a more prominent and consistent progression pattern. This helps explain why trajectory-based confidence changes around the bottleneck provide a clean

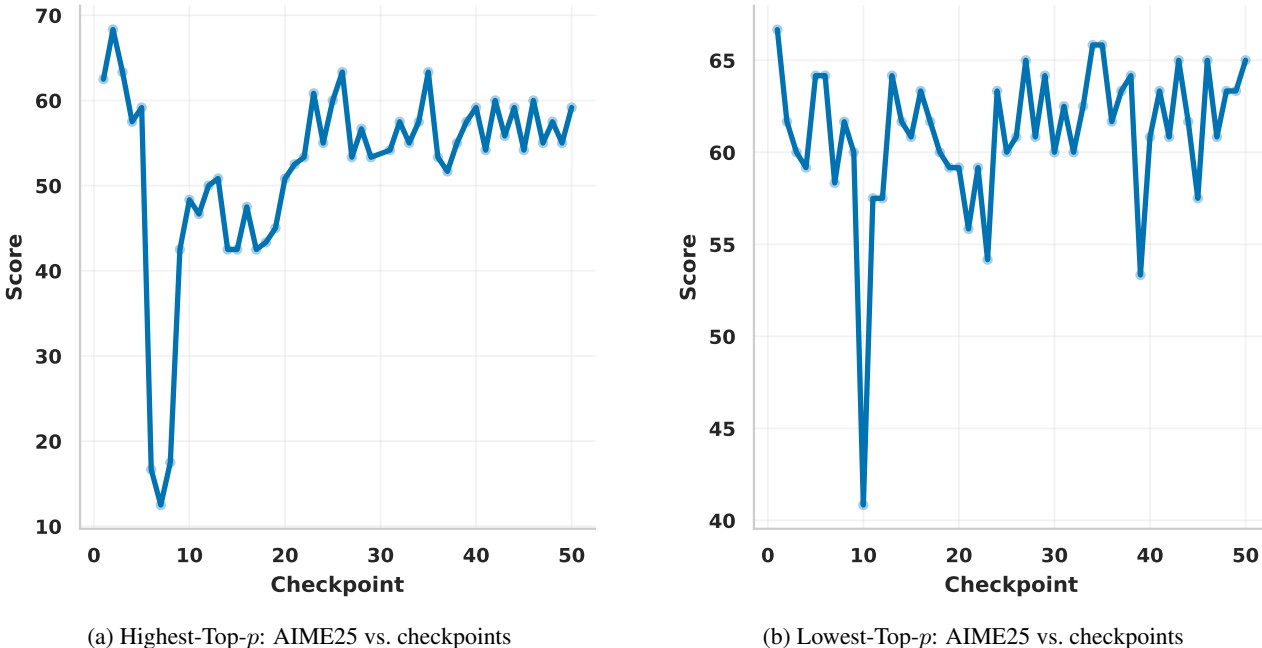

(a) Highest-Top-$p$: AIME25 vs. checkpoints          (b) Lowest-Top-$p$: AIME25 vs. checkpoints

*Figure 13.* **Imitation Shock persists under initial-confidence masking.** Even when masking the same token fraction as T3S, selecting tokens by base-model confidence (highest or lowest) still exhibits the characteristic crash-then-recover pattern in AIME25 across checkpoints.

targeting signal, whereas single-checkpoint heuristics (e.g., initial confidence or local gradient similarity) do not cleanly separate the groups (Appendix Sections J.2 and J.1).

**Summary.** Overall, the loss-transfer analysis provides controlled evidence that (i) the T3S partition corresponds to a meaningful functional separation, and (ii) anchor gradients can exert disproportionate suppressive influence on the remaining tokens. This motivates the central intervention in T3S: **masking anchors early to clear an unobstructed optimization path for the yet-to-learn tokens**.

