# OpenReview forum: "Training-Trajectory-Aware Token Selection"
_ICML.cc/2026/Conference — ICML 2026 regular_

### Official Review · Reviewer_eL1o · 2026-02-18

**Soundness:** 3
**Presentation:** 3
**Significance:** 3
**Originality:** 3
**Overall Recommendation:** 4
**Confidence:** 3

**Summary:**

The paper identifies a training-trajectory phenomenon in continual reasoning distillation, termed _Imitation Shock_, showing performance temporarily collapses despite decreasing training loss. It proposes Training-Trajectory-Aware Token Selection (T3S), which uses token-confidence changes around the bottleneck to mask early-learned “anchor tokens” in autoregressive distillation and prioritize yet-to-learn tokens in diffusion LLM training. The method is evaluated on multiple student–teacher settings and small-budget distillation scenarios, showing improvements over standard SFT and a residual-transfer baseline.

**Compliance With Llm Reviewing Policy:**

Affirmed.

**Final Justification:**

The paper identifies an interesting and reproducible failure mode in continual reasoning distillation and proposes a simple, effective intervention. The rebuttal resolved most of my practical concerns, but I still have some reservations about how fully the causal mechanism is established and about the role of the selector model. Overall, the weaknesses now look more like limitations than major flaws, so I keep my final recommendation at 4.

**Key Questions For Authors:**

1. How sensitive is T3S to errors in bottleneck detection?
For example, what happens if the bottleneck is identified a few checkpoints earlier or later than the true minimum?

2. What fraction of tokens are typically masked by the rule $\Delta c_t > 0 across datasets and models?

3. Can the authors report the computational overhead of trajectory logging and token-confidence computation compared to standard SFT?

4. How sensitive are results to the choice of selector model M0?

5. Can the authors include a direct comparison in the main text with static token-selection baselines (e.g., entropy-based masking)?

**Limitations:**

yes

**Strengths And Weaknesses:**

# Strengths

- Identifies a reproducible training-trajectory failure mode in continual
  reasoning distillation.
- Provides a token-level mechanism explaining the phenomenon.
- Proposes a simple trajectory-aware masking strategy with consistent gains.
- Demonstrates applicability to both AR and diffusion language models.


# Major Weaknesses

- Bottleneck detection relies on training accuracy.

> "We define the Imitation Bottleneck checkpoint as ... the checkpoint with the lowest training accuracy along the distillation trajectory."

---

- Anchor-token masking rule has no explicit threshold or budget control.

> "Operationally, we define A(x, y) = {t ∈ [T] | Δc_t(x, y) > 0}."

---

- Causal evidence is limited to one-step interventions.

> "For each saved checkpoint θ^(k), we take one gradient step that optimizes only Imitation-Anchor Tokens..."

---

- Token-selection baselines are not included in the main results.

> "Recent token-selection approaches often rely on static signals such as initial uncertainty (token entropy)..."

---

- Practical overhead is not quantified.

> "In practice, T3S measurements can be performed online during training."

---

- Role of the selector model M0 is not fully clear.

> "We use an AR token selector model M0 to score token-wise confidence along the training trajectory."

---

- Generality beyond math reasoning distillation is limited in the main experiments.

> "We use two high-quality open-source mathematical reasoning datasets for training..."



# Minor weakness
Typos such as:

**1. Inconsistent method name in Abstract ("T3" vs "T3S").**

> "To this end, we propose Training-Trajectory-Aware Token Selection (T3S) ...
> T3 yields consistent gains in both AR and dLLM settings..."

The method is defined as **T3S**, but later referred to as **T3**.

**2. typo Figures**
>only after the bottleneck do both groups start improving (Figures 4).

>this suppression weakens only after anchor tokens have been learned to a sufficient extent (Figures 5).

**3. Reference Formatting Issues**
> deepseek ai. R1-distill-llama-8b.

>Math-AI. Aime 2024.

>typhoon ai. rl-code-math-v5.

**4. Clause's capital letter**
> ...recovers; We term this phenomenon Imitation Shock.

---

> ### Author Rebuttal · Authors · 2026-03-31
>
> Thank you for the constructive feedback and recognition. We address your questions (Q) and Weaknesses (W) below.
>
> ### W 1/6, Q 1/4
> **Reliance on Training Accuracy:** Actually, training accuracy is the most direct and applicable metric for observing the bottleneck. And we emphasize in Section 3.5 that this requirement is already satisfied by most modern distillation/RL pipelines. Any reasoning dataset with outcome-based verification directly provides this metric.
>
> **Sensitivity:** To test the sensitivity of T3S to errors in bottleneck detection (or the choice of $M_0$), we uniformly sampled checkpoints (ckt) before and after the Real Imitation Bottleneck to serve as $M_0$ and evaluated the final AIME scores:
> * Base Model: 71.46 | SFT: 63.13
> * T3S-ckt 2: 68.34
> * T3S-ckt 4: 72.50
> * T3S-ckt 6: 74.52
> * T3S-ckt 8 (Real Bottleneck): 77.30
> * T3S-ckt 10: 74.42
> * T3S-ckt 20: 73.17
> * T3S-ckt 30: 71.25
> * T3S-ckt 50: 69.48
>
> **Key Findings:**
> 1. The performance forms a clear inverted U-shape peaking at the true bottleneck (ckt 8), mirroring the exact Imitation Shock dynamics described in Figures 1 & 2.
> 2. Remarkably, using even the very first (ckt 2) or the very last (ckt 50) checkpoint as the selector still outperforms direct SFT. This aligns with our analyses in Figure 4/5 and Table 2: imitation-anchor tokens dominate optimization from the very beginning, and a large portion of yet-to-learn tokens remains suppressed even at the end of training. Consequently, both early and late selectors can still successfully target and mask a sufficient number of harmful anchor tokens.
> 3. This proves T3S is highly robust to detection errors and supports fast, heuristic alternatives in practice.
> ### W2, Q2
> T3S does have a strict, data-driven explicit threshold: $\Delta c_t>0$. This is not an arbitrary hyperparameter but a direct outcome of our empirical observations in Section 2. We identified that tokens with a confidence increase ($>0$) relative to the base model by the bottleneck stage are precisely the "imitation anchors", which need to be masked to clear the optimization path for yet-to-learn tokens. Regarding the masked fraction, We empirically find a consistent 20% masking rate for identified anchor tokens across settings.
> ### W3
> Actually, the one-step intervention is not an isolated observation; it is a systematic causal probe [1,2] applied across all saved checkpoints. It reliably demonstrates that optimizing anchors consistently spikes the loss of yet-to-learn tokens. Moreover, we respectfully highlight that the entirety of Section 2 constructs a compounding, multifaceted body of evidence.
> The token-wise learning dynamics (Figure 4) show continuous suppression. The severe performance collapse of the inverted mask ablation (-T3S, Table 3) also proves the exact necessity of this specific partition; and finally, T3S successfully preventing the performance crash during training (Figure 7) closes the loop. Together, these localized probes, macroscopic dynamics, and successful system-level resolutions form a complete and rigorously validated causal chain.
> ### W4, Q5
> Thank you for emphasizing this. We completely agree that the comparison against static baselines is a crucial contribution. We currently detail in Appendices I/J why static signals fail to identify anchors and mitigate Imitation Shock. Specifically, we want to emphasize that static masking based on either high-entropy or low-entropy completely fails to eliminate the Imitation Shock phenomenon, and their final performance lags far behind our proposed T3S. We will promote these static token-selection comparisons to the main text to highlight the necessity of trajectory-aware selection.
> ### W5, Q3
> Because T3S requires logging the trajectory to compute the mask before the final targeted training, the overall wall-clock time is slightly more than 2x that of standard SFT. However, we argue this overhead is highly justified:
> 1. As shown in Table 7, the token selector is transferable across different student models within the same family. This grants a "compute-once, reuse-anywhere" advantage.
> 2. Under a strictly fixed data budget, T3S provides a significantly better way to utilize computation. By investing additional compute into token-level trajectory analysis, our method successfully unlocks the deeper latent potential of the limited training data, ultimately yielding substantially better reasoning performance than what standard SFT could ever achieve on the exact same dataset.
> ### W6
> On a purely code-based dataset, we also observed the Imitation Shock (Figure 2e). Applying T3S to this code dataset also yields similar significant improvements on the LiveCodeBench:
> * Base: 55.76
> * SFT: 53.29
> * T3S: 61.41
> ### Minor Weaknesses
> We sincerely apologize for the typos. They will be thoroughly corrected in the revised manuscript.
>
> ---
>
> [1] Understanding Black-box Predictions via Influence Functions
>
> [2] An Empirical Study of Example Forgetting during Deep Neural Network Learning

---

> > ### Author Rebuttal · Reviewer_eL1o · 2026-04-02
> >
> > The rebuttal addresses most of my practical concerns. The added sensitivity results for bottleneck selection, the clarification of the masking rule and masked fraction, the explicit overhead estimate, and the commitment to move static token-selection baselines into the main text all help.
> >
> > I still think some limitations remain. The causal evidence is still mostly built around one-step interventions plus supporting trends, so I do not think the mechanism is fully nailed down. The role of the selector model is also clearer than before, but not completely clean.
> >
> > That said, the main weaknesses now look more like limitations than major flaws. I keep my score at 4.

---

> > > ### Author Response · Authors · 2026-04-03
> > >
> > > We sincerely thank you for your prompt follow-up and for acknowledging that our rebuttal addressed your practical concerns. We deeply appreciate your continued engagement with our work.
> > >
> > > As discussed, we will thoroughly incorporate the newly added selector sensitivity experiments into the revised version, accompanied by a much clearer and more precise discussion regarding the role of the selector model to ensure there is no ambiguity.
> > >
> > > Regarding your remaining concern about the causal evidence, to provide a more fundamental explanation and to directly expose the optimization landscape, we have conducted an additional formal analysis that tracks the gradient dynamics between the two token groups at every checkpoint. This deeper look into the optimization process better supports our claims:
> > >
> > > **A. Gradient Norm:**
> > > In the initial training steps, the gradient norm of Anchor tokens is overwhelmingly larger than that of Other tokens (up to 17x). This massive magnitude disparity dictates the early optimization process. As Anchor loss rapidly drops (corresponding to Figure 5b), their gradient norm sharply contracts. By the Imitation Bottleneck, the ratio drops to approximately 2x. It is only at this point that the gradients of Other tokens may gain a sufficient "voice" in the optimization update, allowing downstream performance to recover. Crucially, the gradient norm of Anchor tokens remains larger than that of Other tokens throughout the entire training process. Without masking, Anchors permanently dominate the optimization bandwidth.
> > >
> > > **B. Gradient Directions:**
> > > To test if the two groups share benign or conflicting optimization directions, we computed the cosine similarity between their gradients. During the initial crash phase, the similarity drops from -0.4 to -0.5, indicating severe directional conflict. Right after the bottleneck, during the rapid recovery phase, the similarity briefly spikes to -0.1, before settling back to -0.4. However, the similarity is **consistently negative** throughout training, which provides another solid proof for our observation in Table 5: the optimization landscapes of these two token groups are antagonistic, and progress on one inherently opposes the other.
> > >
> > > Detailed visualizations of these dynamics have been provided in the anonymous link [1]. Building upon these new results, we will also include a formal clearer discussion in the revised manuscript.
> > >
> > > We hope these deeper mechanistic insights alleviate your remaining concerns. Please let us know if you have any lingering questions or specific nuances you feel we should further clarify. We are fully committed to completely addressing this point and would be more than happy to engage in further discussion if any doubts remain. Thank you again for pushing us to strengthen the foundational explanation of this phenomenon!
> > >
> > > [1] https://anonymous.4open.science/r/T3S_ICML-B66F/t3s_grad_analysis.png

---

### Official Review · Reviewer_CFXi · 2026-03-07

**Soundness:** 3
**Presentation:** 3
**Significance:** 3
**Originality:** 4
**Overall Recommendation:** 4
**Confidence:** 2

**Summary:**

This paper investigates the failure mode of continual reasoning distillation when the student model already possesses strong reasoning capabilities. The authors identify a phenomenon called "Imitation Shock," where performance metrics crash to a consistent minimum (the "Imitation Bottleneck") and then recover, even as training loss decreases monotonically. Through token-level analysis, the paper reveals that "Imitation-Anchor Tokens" dominate early optimization and suppress learning of other beneficial tokens. Based on this mechanistic insight, the authors propose Training-Trajectory-Aware Token Selection (T3S), which masks anchor tokens to clear the optimization path for yet-to-learn tokens. T3S achieves consistent gains in both autoregressive and diffusion language model settings, enabling Qwen3-8B to surpass DeepSeek-R1 and achieving SOTA among 16B-scale no-think models for dLLMs with only hundreds of training examples.

**Compliance With Llm Reviewing Policy:**

Affirmed.

**Key Questions For Authors:**

Can you provide a more formal analysis of why Imitation-Anchor Tokens suppress yet-to-learn tokens? For example, is there a gradient interference analysis or optimization landscape argument that explains this coupling? A positive answer would significantly strengthen the theoretical foundation.

How sensitive is T3S to the exact identification of the Imitation Bottleneck?

Have you experimented with softer token weighting schemes instead of the binary mask?

What is the actual wall-clock overhead of T3S compared to standard SFT?

**Limitations:**

The authors have partially discussed limitations, noting the dependence on training accuracy and verifier access.

**Strengths And Weaknesses:**

The discovery of Imitation Shock and its token-level explanation is compelling. The paper provides multiple complementary pieces of evidence—training dynamics analysis, one-step interventions (Figure 5), the Recovering-Residual Transfer (RRT) construction, and a 4x4 loss-transfer asymmetry matrix (Table 5)—that together build a convincing causal story. This level of mechanistic understanding is rare in distillation literature.

The comparison between T3S and its inverse (-T3S) is a strong diagnostic that demonstrates T3S identifies a meaningful and selective token partition rather than merely discarding redundant positions. The severe performance collapse under -T3S confirms the token selection is essential.

Weaknesses:

While the empirical analysis is thorough, the paper does not provide a formal theoretical explanation for why Imitation-Anchor Tokens suppress other tokens. A formal analysis (e.g., through gradient dynamics or optimization landscape arguments) would strengthen the contribution.

The paper mentions T3S can be performed "online during training" with "minimal overhead," but does not provide concrete wall-clock comparisons or FLOPs analysis. The need to log and evaluate every checkpoint could be non-trivial in practice.

---

> ### Author Rebuttal · Authors · 2026-03-31
>
> Thank you for highly commending our insights! Below, we address your questions point-by-point.
> ### Q1
> While our primary discoveries and conclusions are empirical and grounded in extensive experimental observations, we completely agree that analyzing the gradient dynamics better supports our claims and tracks the gradient norms and directions between two token groups at every checkpoint:
>
> **A. Norm:**
> In the initial training steps, the gradient norm of Anchor tokens is overwhelmingly larger than that of Other tokens (up to 17x). This massive magnitude disparity dictates the early optimization process. As Anchor loss rapidly drops (corresponding to Figure 5b), their gradient norm sharply contracts. By the Imitation Bottleneck, the ratio drops to approximately 2x. It is only at this point that the gradients of Other tokens may gain a sufficient "voice" in the optimization update, allowing downstream performance to recover. Crucially, the gradient norm of Anchor tokens remains larger than that of Other tokens throughout the entire training process. Without masking, Anchors permanently dominate the optimization bandwidth.
>
> **B. Directions:**
> To test if the two groups share benign or conflicting optimization directions, we computed the cosine similarity between their gradients. During the initial crash phase, the similarity drops from -0.4 to -0.5, indicating severe directional conflict. Right after the bottleneck, during the rapid recovery phase, the similarity briefly spikes to -0.1, before settling back to -0.4. However, the similarity is **consistently negative** throughout training, which provides another proof for our observation in Table 5: the optimization landscapes of these two token groups are antagonistic, and progress on one inherently opposes the other.
>
> Detailed visualizations have been provided in the anonymous link [1]. Building upon these new results, we will include a formal theoretical discussion in the revised manuscript.
>
> [1] https://anonymous.4open.science/r/T3S_ICML-B66F/t3s_grad_analysis.png
> ### Q2
> To test the sensitivity of T3S in bottleneck detection, we uniformly sampled checkpoints (ckt) before and after the Real Imitation Bottleneck to serve as the selector and evaluated the final scores:
> * Base Model: 71.46 | SFT: 63.13
> * T3S-ckt 2: 68.34
> * T3S-ckt 4: 72.50
> * T3S-ckt 6: 74.52
> * T3S-ckt 8 (Real Bottleneck): 77.30
> * T3S-ckt 10: 74.42
> * T3S-ckt 20: 73.17
> * T3S-ckt 30: 71.25
> * T3S-ckt 50: 69.48
>
> **Key Findings:**
> 1. The performance forms a clear inverted U-shape peaking at the true bottleneck (ckt 8), mirroring the exact Imitation Shock dynamics described in Figures 1 & 2.
> 2. Using even the very first (ckt 2) or the very last (ckt 50) checkpoint as the selector still outperforms direct SFT. This aligns with our analyses in Figure 4/5 and Table 2: imitation-anchor tokens dominate optimization from the very beginning, and a large portion of yet-to-learn tokens remains suppressed even at the end of training. Consequently, both early and late selectors can still successfully target and mask a sufficient number of harmful anchor tokens.
> 3. This proves T3S is highly robust to detection errors and supports fast, heuristic alternatives in practice.
>
> ### Q3
> We opted for a strict binary mask rather than a softer scheme for two primary reasons:
> 1. As demonstrated by the consistently negative gradient cosine similarity discussed above (and the loss-transfer matrix in Table 5), two token groups form a highly separable, antagonistic partition. Their gradients do not coexist benignly. Therefore, when progress on one side explicitly comes at the expense of the other, applying a hard binary mask is the most intuitive and direct intervention to cleanly sever this interference.
> 2. Soft weighting introduces a massive hyperparameter search space. In contrast, the binary mask driven by $\Delta c_t > 0$ is a parameter-free, data-driven boundary that is empirically highly robust across different models, datasets, and domains. However, given the strong performance of the robust binary mask, we believe exploring complex, high-dimensional continuous weighting schemes in the optimization landscape also remains a highly promising avenue for our future work.
>
> ### Q4
> Because T3S requires logging the trajectory to compute the mask before the final targeted training, the overall wall-clock time is slightly more than 2x that of standard SFT. However, we argue it is highly justified:
> 1. As shown in Table 7, the token selector is transferable across different student models within the same family. This grants a "compute-once, reuse-anywhere" advantage.
> 2. Under a strictly fixed data budget, T3S provides a much better way to utilize computation. By investing additional compute into token-level analysis, our method successfully unlocks the deeper latent potential of the limited training data, ultimately yielding substantially better reasoning performance than what standard SFT could ever achieve on the exact same dataset.

---

> > ### Author Rebuttal · Reviewer_CFXi · 2026-04-02
> >
> > Thanks for the rebuttal. I think the author has addressed my concerns.

---

> > > ### Author Response · Authors · 2026-04-02
> > >
> > > Dear Reviewer,
> > >
> > > We are very grateful for your prompt confirmation and are thrilled to hear that our rebuttal successfully addressed your concerns! Your insightful suggestion to analyze the gradient dynamics significantly elevated the depth of our work, and we look forward to incorporating these new findings into the final manuscript.
> > >
> > > Best regards,
> > >
> > > The Authors

---

### Official Review · Reviewer_CNmD · 2026-03-12

**Soundness:** 2
**Presentation:** 3
**Significance:** 3
**Originality:** 3
**Overall Recommendation:** 4
**Confidence:** 4

**Summary:**

This paper finds that during large language model distillation, performance metrics often first decline and then improve. The authors attribute this phenomenon to the negative effect of “imitation-anchored tokens.” Through experiments, they show that confidence gradually separates tokens into two categories. The authors argue that these two types of tokens cannot effectively coexist, and that the first type often hinders the gains brought by the second. Based on this observation, they propose the T3S method, which masks out the first type of token during training to reduce the negative impact of “imitation-anchored tokens” and concentrate computational resources on the second type of tokens that can bring positive learning benefits.

**Compliance With Llm Reviewing Policy:**

Affirmed.

**Key Questions For Authors:**

See the weakness.

**Strengths And Weaknesses:**

Strength:

* The authors identify an insightful training phenomenon and provide a plausible mechanistic explanation. Specifically, they show that during model distillation, even when the training loss decreases monotonically, the evaluation performance may first decline and then improve; they further explain this behavior from a token-level perspective, arguing that there exist two incompatible types of tokens, where one harms training in the early stage and delays the gains brought by the other until later in training.
* The paper is well written and easy to understand. Its presentation is clear and well-organized, making the main ideas easy to follow.
* The authors conduct experiments on both autoregressive and diffusion models, demonstrating the scalability of the T3S method.

Weakness:
1. The experimental evaluation is insufficient:

* The main experiments only compare standard distillation with several variants of T3S. However, some other distillation strategies have also identified similar issues, attributing them to the distribution mismatch between the teacher and student models in the early stage of training, and have proposed several corresponding methods [1,2, 3].

* The authors are encouraged to include these methods in the comparison as well. In addition, the experiments only use DeepSeek-R1 and QWQ-32B as teacher models, and Qwen3-8B and Qwen3-32B as student models. It would strengthen the paper to evaluate the method on a broader range of models and parameter scales, for example, by including Qwen3-235B-Thinking as a teacher model and Llama3 8B as a student model.

* Finally, the experiments are conducted only on the BOBA and S1K datasets. The authors are encouraged to further validate the method on more datasets, such as the OpenThoughts series or LIMO.

[1] Agarwal R, Vieillard N, Zhou Y, et al. On-policy distillation of language models: Learning from self-generated mistakes[C]//The twelfth international conference on learning representations. 2024.
[2] Koo J, Hwang Y, Kim Y, et al. SWITCH: Studying with Teacher for Knowledge Distillation of Large Language Models[C]//Findings of the Association for Computational Linguistics: NAACL 2025. 2025: 3733-3746.
[3] Li Y, Yue X, Xu Z, et al. Small models struggle to learn from strong reasoners[C]//Findings of the Association for Computational Linguistics: ACL 2025. 2025: 25366-25394.

2. Efficiency and computational overhead:

* Although the authors claim that T3S does not introduce much additional overhead, the paper does not clearly explain how the bottleneck is estimated, or whether training needs to be restarted from scratch after the bottleneck is identified. The authors are encouraged to include an experiment on time/computational cost to better demonstrate the practical advantage of T3S.

3. Lack of detailed analysis.

* The authors distinguish between the two types of tokens based on the trajectory of confidence changes during training, but they do not provide sufficient analysis of the semantic properties of these two token groups. Although a word cloud is presented, it does not lead to a clear conclusion. The authors are encouraged to further analyze the semantic correlation or distinction between these two types of tokens. In addition, beyond these two groups with clear trends, there may also exist tokens whose trajectories are less pronounced. It would be helpful if the authors could analyze this possibility and report the proportions of the two token types (as well as a possible third type) in the training data. Furthermore, the paper currently reports only benchmark accuracy. It would strengthen the evaluation to include some measures of language quality, such as the perplexity of the reasoning chains or the self-consistency of the reasoning process.

4. Small things:
* Regarding Figure 2(c), it seems that on the larger-scale dataset, the model accuracy does not clearly exhibit the “first decreases and then increases” trend claimed by the authors.

---

> ### Author Rebuttal · Authors · 2026-03-31
>
> We sincerely thank you for for acknowledging the insightfulness and clarity of our work. We address your concerns point-by-point below.
>
> ### **Weakness 1**
> **A. Comparison with Existing Distillation Strategies**
> Thank you for pointing out these important related works. We have discussed and evaluated them under our continual reasoning distillation setting.
> * **GKD [1] & SWITCH [2]:** A fundamental limitation of them is that they require the teacher model to generate token-level information. This not only introduces massive computational overhead and requires the teacher to be locally deployable, but it also strictly demands that the **teacher and student share the exact same tokenizer**. This completely breaks our core setting: distilling from a powerful Out-of-Distribution teacher like DeepSeek-R1 to Qwen3.
> Nevertheless, we also evaluated them in an tokenizer-sharing setting (Teacher: Qwen3-32B, Student: Qwen3-8B). As shown below, T3S significantly outperforms both:
> | Method | AIME24 | AIME25 |
> | :--- | :--- | :--- |
> | GKD [1] | 73.33 | 53.33 |
> | SWITCH [2] | 75.83 | 66.67 |
> | T3S (Ours) | 80.83 | 70.00 |
> * **Mix Distillation [3]:** We also compared T3S against Mix Distillation under Two different types of large teachers and small teachers' combinations (R1/QWQ and Qwen3-235B/QWQ). T3S consistently achieves much higher average scores (AIME24/25):
> | teachers' combinations | Mix Distillation | T3S |
> | :--- | :--- | :--- |
> | R1/QWQ | 68.75 | 75.84 |
> | Qwen3-235B/QWQ | 71.67 | 77.40 |
>
> **B. Evaluation on Broader Models**
> We applied T3S to a completely different student architecture, using Qwen3-235B to continually distill R1-Distill-Llama3-8B. The results confirm that T3S remains highly effective across different model families:
> |  | AIME24 | AIME25 |
> | :--- | :--- | :--- |
> | Base model | 45.00 | 33.33 |
> | SFT | 50.00 | 36.67 |
> | T3S | 56.04 | 55.83 |
>
> **C. Validation on More Datasets**
> We further expanded our evaluation to the datasets you recommended: a 3K sample subset of OpenThoughts-3 and the full LIMO dataset, using DeepSeek-R1 as the teacher. T3S strictly dominates standard SFT in these diverse and larger-scale settings:
> | Dataset | SFT | T3S |
> | :--- | :--- | :--- |
> | OpenThoughts-3 (3K) | 52.50 | 73.13 |
> | LIMO (Full) | 64.58 | 78.33 |
>
> ### **Weakness 2**
>
> Because T3S requires logging the trajectory to compute the mask before the final targeted training, the overall wall-clock time is slightly more than 2x that of standard SFT. However, we argue this overhead is highly justified:
> 1. As shown in Table 7, the token selector is transferable across different student models within the same family. This grants a "compute-once, reuse-anywhere" advantage.
> 2. Under a strictly fixed data budget, T3S provides a significantly better way to utilize computation. By investing additional compute into token-level trajectory analysis, our method successfully unlocks the deeper latent potential of the limited training data, ultimately yielding substantially better reasoning performance than what standard SFT could ever achieve on the exact same dataset.
>
> ### **Weakness 3**
> **Semantic Properties:** This is an insightful question. We conducted a deeper linguistic analysis and found that "imitation anchors" disproportionately consist of highly expressive tokens (e.g., specific nouns and adjectives). Because these tokens may have more valid synonyms, perhaps it is more likely for these tokens to have the situation where the teacher's output tokens deviate from the expression distribution of the student model. We explicitly verified this by computing the Teacher-Student KL divergence, which is significantly higher on anchor tokens than on others. We hypothesize that forcing the student to overfit these stylistic/expressive choices early on derails the learning of the actual reasoning logic.
>
> **Token Proportions:** We evaluated the stability of this partition by running 5 independent training runs with random data shuffling. The proportion consistently stabilized at an approximate 20% (Anchors) to 80% (Others) split. Only Less than 1% of tokens fluctuated between categories.
>
> **Language Quality & Consistency:** 1. Regarding perplexity, we have implicitly addressed this through our training loss analysis (Figure 7, a sample-level proxy) and token confidence discussions (Section 4.4, a token-level proxy).
> 2. Regarding reasoning self-consistency, the consistency of the reasoning process is reflected in our outcome model. Across 5 repeated evaluations, T3S yielded an AIME average of 76.98 ± 0.35 on BOBA-200 and 76.61 ± 0.38 on S1K-200. This low variance demonstrates that the reasoning chains generated by T3S are highly consistent and robust.
>
> ### **Weakness 4**
>
> In Figure 2(c), the model's accuracy starts near 60, drops sharply to approximately 35 (the clear global minimum bottleneck), and then steadily recovers to the 45-50 range. The "crash and recover" trajectory remains distinctly present even on the larger-scale dataset.

---

> > ### Author Rebuttal · Reviewer_CNmD · 2026-04-01
> >
> > Thanks for the author's rebuttal. I think the author has addressed most of my concerns. I will adjust my score.

---

> > > ### Author Response · Authors · 2026-04-01
> > >
> > > Dear Reviewer,
> > >
> > > We sincerely thank you for your clear and constructive feedback. Your suggestions directly motivated our new experiments, which have greatly improved the quality of our paper. We are thrilled that our rebuttal resolved your concerns and sincerely appreciate the score increase.
> > >
> > > Best regards,
> > >
> > > The Authors

---

### Official Review · Reviewer_Pdih · 2026-03-12

**Soundness:** 3
**Presentation:** 4
**Significance:** 3
**Originality:** 3
**Overall Recommendation:** 5
**Confidence:** 3

**Summary:**

The authors observe a phenomenon that model performance generally drops sharply before gradually recovering during the distillation period. They further uncover that certain imitation-anchor tokens quickly anchor optimization and can exert a strong suppressive effect on other tokens. As a result, the authors propose the T3S (Training-Trajectory-Aware Token Seletion) method. Specifically, T3S discriminates anchor tokens through confidence difference and prevent the model from learning from these tokens. Extensive experiments demonstrate the effectiveness of T3S.

**Compliance With Llm Reviewing Policy:**

Affirmed.

**Key Questions For Authors:**

Please refer to the Cons part.

1. Please consider conducting corresponding ablation studies.
2. Please consider conducting experiments on larger dataset scales.

**Limitations:**

yes

**Strengths And Weaknesses:**

Pros:

1. The paper is well-written and easy to follow.
2. The introduced Imitation Shock is a general phenomenon in various scenarios, and novel to be investigated.
3. Sufficient experiments claify the motivation and effectiveness of T3S.

Cons:

1. The current implementation of T3S relies on training accuracy to identify the Imitation Bottleneck. However, this explicit metric can be unavailable for open-ended or subjective tasks like creative writing. Accoring to Figure1 and Figure2, the imitation bottleneck appears to consistently occur around a specific training stage. I wonder if such a simple empirical heuristic could achieve comparable performance. I suggest the authors provide corresponding ablation studies on this heuristic strategy.

2. Current experiments mainly focus on evaluation on tiny datasets (for instance BOBA-200 and S1K-200). Although the authors demonstrate that Imitation Shock happens under larger dataset scales in Appendix E, there is no corresponding SFT experiments that demonstrates the effectiveness of T3S under such training scale.

---

> ### Author Rebuttal · Authors · 2026-03-31
>
> We sincerely thank you for the positive evaluation and for recognizing the novelty of our paper. We highly appreciate your constructive suggestions. Below are our detailed responses.
>
> ---
>
> ### **Response to Weakness 1 & Question 1: Empirical Heuristic and Ablation Study**
>
> We agree with your assessment. Our current work primarily focuses on reasoning capabilities, where objective metrics like training accuracy are readily available. For subjective tasks like creative writing, extending T3S by utilizing emerging automated evaluation methods, such as LLM-as-a-judge with specific rubrics [1,2], is a highly promising direction for our future work.
>
> More importantly, **your observation that the imitation bottleneck consistently occurs around a specific training stage is brilliantly insightful.** Following your suggestion, we closely examined the occurrence of the bottleneck and found it consistently falls between **step 6 and step 15**.
>
> To test the viability of a simple empirical heuristic, we implemented **T3S-simple**, which blindly selects the model checkpoint at **step 10** as the token selector, bypassing the need to search for the exact lowest training accuracy. As shown in the table below (using Qwen3-8B as the student), while T3S-simple slightly trails the exact T3S, it still significantly outperforms direct SFT:
>
> | Dataset | Method | AIME24 | AIME25 |
> | :--- | :--- | :--- | :--- |
> | BOBA-200 | Base | 75.83 | 67.08 |
> | | SFT (R1) | 71.25 | 55.00 |
> | | T3S-simple (R1) | 78.83 | 70.00 |
> | | T3S (R1) | 80.63 | 73.96 |
> | S1K-200 | Base | 75.83 | 67.08 |
> | | SFT (R1) | 72.50 | 55.83 |
> | | T3S-simple (R1) | 77.50 | 70.21 |
> | | T3S (R1) | 80.00 | 73.13 |
>
> Moreover, to comprehensively explore issues regarding heuristic strategies and the underlying mechanism of T3S, we conducted an extensive ablation study. We uniformly sampled checkpoints (ckt) before and after the Real Imitation Bottleneck to serve as the token selector and evaluated the final AIME 24/25 averaged scores after applying T3S.
>
> * **Base Model:** 71.46 | **SFT:** 63.13
> * **T3S-ckt 2:** 68.34
> * **T3S-ckt 4:** 72.50
> * **T3S-ckt 6:** 74.52
> * **T3S-ckt 8 (Real Imitation Bottleneck): 77.30**
> * **T3S-ckt 10:** 74.42
> * **T3S-ckt 20:** 73.17
> * **T3S-ckt 30:** 71.25
> * **T3S-ckt 40:** 70.11
> * **T3S-ckt 50:** 69.48
>
> **Key Takeaways:**
> 1. The performance trend exhibits a clear inverted U-shape, peaking exactly at the real imitation bottleneck (ckt 8). This perfectly mirrors the "crash then recover" dynamic of the Imitation Shock phenomenon described in Figures 1 & 2, further validating our mechanism design.
> 2. Remarkably, using even the very first (ckt 2) or the very last (ckt 50) checkpoint as the selector still outperforms direct SFT. This aligns with our analyses in Figure 4/5 and Table 2: imitation-anchor tokens dominate optimization from the very beginning, and a large portion of yet-to-learn tokens remains suppressed even at the end of training. Consequently, both early and late selectors can still successfully target and mask a sufficient number of harmful anchor tokens.
> 3. T3S maintains a positive gain over a remarkably wide range of checkpoints. This solidifies your hypothesis: fast and convenient heuristic strategies (like T3S-simple) are highly viable and robust in practice.
>
> ---
>
> ### **Response to Weakness 2 & Question 2: Experiments on Larger Dataset Scales**
>
> Thank you for prompting us to demonstrate the scalability and generality of T3S. We have conducted two new sets of experiments scaling up the distillation data significantly:
> 1.  DeepSeek-R1 distilling 3K samples from OpenThought-3.
> 2.  Qwen3-235B distilling the full BOBA dataset (**~100K** samples).
>
> | Dataset (Scale) | Method | AIME24 | AIME25 |
> | :--- | :--- | :--- | :--- |
> | **OpenThought-3 (3K)** | Base | 75.83 | 67.08 |
> | Teacher: DeepSeek-R1 | SFT | 60.00 | 45.00 |
> | | **T3S** | **77.50** | **68.64** |
> | **BOBA (~100K)** | Base | 75.83 | 67.08 |
> | Teacher: Qwen3-235B | SFT | 75.83 | 70.00 |
> | | **T3S** | **80.21** | **72.71** |
>
> These results demonstrate that T3S scales effectively. When learning from an out-of-distribution teacher (DeepSeek-R1), direct continual distillation still leads to severe degradation. T3S completely prevents this collapse and manages to extract beneficial signals, yielding net positive gains. Furthermore, when learning from an in-distribution teacher at a massive scale, T3S continues to amplify the learning gains far beyond what standard SFT achieves.
>
> ---
>
> We once again express our sincere gratitude for your insightful review and constructive suggestions. Your feedback has genuinely helped us strengthen the empirical foundation and practical value of our work. We will carefully incorporate all these discussions into the final revised manuscript. We hope these additions fully address your remaining questions.
>
> ---
>
> [1] Reinforcement learning with rubric anchors
>
> [2] Litbench: A benchmark and dataset for reliable evaluation of creative writing

---

> > ### Author Rebuttal · Reviewer_Pdih · 2026-04-01
> >
> > My major concerns have been explained. I will keep my positive score.

---

> > > ### Author Response · Authors · 2026-04-01
> > >
> > > Dear Reviewer,
> > >
> > > Thank you for taking the time to review our rebuttal and for confirming that your concerns have been resolved. We deeply appreciate your continued support and the highly constructive feedback you have provided, which has undoubtedly helped us strengthen the final version of our paper.
> > >
> > > Best regards,
> > >
> > > The Authors

---

### Decision · Program_Chairs · 2026-04-30

**Decision:**

Accept (regular)

**Comment:**

This paper identifies "Imitation Shock," a reproducible crash-then-recover pattern during continual reasoning distillation, and proposes T3S to address it by masking "Imitation-Anchor Tokens" whose rapid confidence growth suppresses learning of harder tokens (i.e., it's much easier to learn those tokens than actually more useful tokens). The method extends to diffusion LLMs, with strong results on AIME24/25 using a small number of training examples.

The rebuttal effectively addressed the main concerns, with new baselines, large-scale experiments, and gradient dynamics (both norm and directions) analysis resolving most technical questions. The identification of Imitation Shock as a previously uncharacterized phenomenon, combined with a principled mechanistic fix, represents a genuine contribution. Some questions about the completeness of the causal mechanism remain, but the overall strength of the contribution justifies acceptance.